# A unified view of low complexity regions (LCRs) across species

**Byron Lee[1†], Nima Jaberi-Lashkari[1†], Eliezer Calo[1,2]***

[1]Department of Biology, Massachusetts Institute of Technology, Cambridge, United States; [2]David H. Koch Institute for Integrative Cancer Research, Massachusetts Institute of Technology, Cambridge, United States

**Abstract** Low complexity regions (LCRs) play a role in a variety of important biological processes, yet we lack a unified view of their sequences, features, relationships, and functions. Here, we use dotplots and dimensionality reduction to systematically define LCR type/copy relationships and create a map of LCR sequence space capable of integrating LCR features and functions. By defining LCR relationships across the proteome, we provide insight into how LCR type and copy number contribute to higher order assemblies, such as the importance of K-rich LCR copy number for assembly of the nucleolar protein RPA43 in vivo and in vitro. With LCR maps, we reveal the underlying structure of LCR sequence space, and relate differential occupancy in this space to the conservation and emergence of higher order assemblies, including the metazoan extracellular matrix and plant cell wall. Together, LCR relationships and maps uncover and identify scaffold-client relationships among E-rich LCR-containing proteins in the nucleolus, and revealed previously undescribed regions of LCR sequence space with signatures of higher order assemblies, including a teleost-specific T/H-rich sequence space. Thus, this unified view of LCRs enables discovery of how LCRs encode higher order assemblies of organisms.

**\*For correspondence:**
calo@mit.edu

[†]These authors contributed equally to this work

**Competing interest:** The authors declare that no competing interests exist.

## Editor's evaluation

This is a valuable paper that discusses a holistic understanding of LCRs of not only individual proteins but also provides a broad perspective of proteomes across many species. The data presented provides solid evidence on how LCR organisation and assemblies may be shared between subcellular compartmentalisation and extracellular organismal structure.

## Introduction

Low complexity regions (LCRs) of proteins, sequences that frequently repeat the same amino acids (e.g. 'AAAAAAA' or 'LLQLLLSLL'), are abundant in proteomes (*DePristo et al., 2006*; *Huntley and Clark, 2007*; *Haerty and Golding, 2010*). Yet, despite their abundance, we only understand the functions of a small fraction of these LCRs. These contiguous regions of low sequence entropy are found in proteins which play a role in many different biological processes such as transcription, stress response, and extracellular structure (*Coletta et al., 2010*; *Cascarina and Ross, 2018*; *Mier et al., 2020*). The role of LCRs in processes across such disparate fields of biology has made it a challenge to understand how LCR features can give rise to these seemingly different functions.

More recently, some LCR-containing proteins have been shown to direct the higher order assembly of intracellular membraneless bodies (*Patel et al., 2015*; *Yang et al., 2020*). In identifying features of LCRs important for these assemblies, work in this field has begun to provide more general insight into how LCRs can encode their disparate functions. Experimental approaches, such as NMR and SAXS, have found examples where specific residues are required for the intermolecular interactions

responsible for higher order assembly (*Kim et al., 2019b*; *Martin et al., 2020*). Computational identification of short linear motifs (SLiMs) have cataloged specific sub-sequences in LCRs which mediate certain interactions and post-translational modifications (*Krystkowiak and Davey, 2017*; *Kumar et al., 2020*), and biophysical predictions of LCRs have given insight into how certain physical properties may direct self-assembly of large compartments (*Das and Pappu, 2013*; *Martin et al., 2020*). Valency, defined by the number of binding sites in a molecule, facilitates the formation of higher order assemblies through interactions between multivalent scaffold proteins, which recruit low-valency clients (*Banani et al., 2016*). Valency can be encoded in any type of sequence (*Li et al., 2012*; *Banani et al., 2016*; *Banani et al., 2017*), yet it has only been studied in a few LCRs.

Numerous proteins have multiple LCRs, and the sequence relationships between these LCRs can impact protein function and higher order assembly. Recent work has shown that in proteins with multiple LCRs, the contributions of individual LCRs on protein function can depend on their identities (*Hebert and Matera, 2000*; *Mitrea et al., 2016*; *Yang et al., 2020*). Synthetic systems have shown that multiple copies of the same LCR can increase the valency of a protein (*Schuster et al., 2018*). However, the extent to which multiple copies of compositionally similar LCRs contribute to valency in natural proteins has not been broadly studied. Furthermore, studies of proteins with compositionally distinct LCRs have shown that they can differentially contribute to the function of the protein, likely through their abilities to interact with different sequences (*Hebert and Matera, 2000*; *Mitrea et al., 2016*; *Yang et al., 2020*). Thus, the copy number and type of LCRs in proteins has large effects on their function for the few types of LCRs where they have been molecularly studied. The fact that copy number and type are not defined by a specific sequence means that they may have more general importance for the functions of LCR-containing proteins.

More broadly, the importance of LCR features and relationships discussed above is not restricted to proteins of intracellular higher order assemblies. Structural assemblies such as the extracellular matrix (*Forgacs et al., 2003*; *Rauscher and Pomès, 2017*), spider silk (*Xu and Lewis, 1990*; *Hinman and Lewis, 1992*; *Malay, 2020*), and the siliceous skeleton of certain sponges (*Shimizu et al., 2015*) are comprised of proteins which share features with proteins involved in intracellular assemblies, such as multivalent scaffolding proteins abundant in LCRs. In fact, many of the proteins comprising these assemblies are composed almost entirely of LCRs which are known to mediate multivalent interactions (*Rauscher et al., 2006*; *Malay, 2020*), suggesting that the contribution of LCRs to valency and hierarchical assembly also play a role in these diverse extracellular assemblies. While the functions of LCRs are disparate, these examples illustrate that many of these functions are a consequence of their ability to assemble. Thus, the features of LCRs discussed above which mediate their higher order assembly may more generally underlie the functions of LCRs across different organisms and biological contexts.

We sought to approach this hypothesis by asking if a holistic approach for studying LCRs which incorporates their sequences, features, and relationships would provide a unified view of LCR function. Given that LCRs are required for such diverse assemblies, how diverse are the sequences of natural LCRs, especially given their low sequence complexity? How do the sequences, biophysical properties, copy number and type of LCRs relate to their roles in the higher order assemblies they form? Can a unified view of LCRs reveal detailed insights into how specific LCRs contribute to higher order assemblies, and provide a broader understanding of LCR sequences and their disparate functions across species?

Here, we use systematic dotplot analysis to provide a comprehensive, unified view of LCRs which spans from single proteins to multiple proteomes. Our approach defined LCR type and copy number across the proteome, allowing us to determine the importance of LCR copy number for the higher order assembly of RPA43 in the nucleolus. Furthermore, using dimensionality reduction, we provide a complete view of LCR sequence space and highlight the continuum of sequences in which natural LCRs exist. We uncover the prevalence of E-rich LCR sequences among human nucleolar proteins and use LCR copy number analyses and LCR maps to identify E-rich LCR-containing proteins which act as scaffolds or clients in the nucleolus. To understand the relationship between LCR sequence and higher order assemblies more broadly, we applied our approach to the proteomes of several species, and found that conservation and emergence of higher order assemblies is reflected in occupancy of LCR sequence space. Together, the principles we learned allowed us to discover a teleost-specific, T/H-rich region of LCR sequence space with signatures of higher order assemblies. Through this unified view, our understanding of LCRs can expand beyond isolated features

or functions, enabling further study of how LCRs, and the higher order assemblies they make up, function in organisms.

## Results
### Dotplots reveal the presence and organization of LCRs in proteins

To gain a view of LCRs and their relationships, we leveraged the dotplot matrix method of sequence comparison (*Gibbs, 1970*; *Pearson and Lipman, 1988*). In self-comparison dotplots, every position in the protein sequence is compared to every other position in the protein in a 2D-matrix. Any position where the two corresponding amino acids are identical is assigned a dot, while non-matching positions are not. Self-comparison dotplots are symmetrical across the diagonal, which represents the comparison of each position in the protein to itself. Within an LCR, the frequent recurrence of amino acids leads to many dots, which appear as a dense square region centered on the diagonal. Moreover, for proteins with multiple LCRs, compositionally similar LCRs will result in dense square regions off of the diagonal in the position corresponding to the intersection of both LCRs, but different LCRs will not intersect in this way. Therefore, dotplots are capable of identifying LCRs in proteins and revealing the relationships between similar and distinct LCRs for proteins with multiple LCRs.

For example, in the dotplot of G3BP1, a protein important for stress granule assembly (*Yang et al., 2020*), the dense squares along the diagonal clearly distinguish between the LCRs of G3BP1 and its other regions, which include its RNA-recognition motif (RRM) and dimerization domains (*Figure 1A*, dotted black outlines). Immediately apparent from the dotplot of G3BP1 is that its LCRs are not all the same type as dense squares do not occupy every intersection off the diagonal (*Figure 1A*, green and red arrows). The dotplot reveals that G3BP1 has LCR relationships where two LCRs are similar to each other (LCR1 and 2) and distinct from a third LCR (LCR3). Previous studies have shown that the presence of these compositionally distinct LCRs are critical for the ability of G3BP1 to form stress granules (*Guillén-Boixet et al., 2020*; *Yang et al., 2020*). The acidic LCRs (LCR1 and 2 in *Figure 1A*) interact with and inhibit the RGG domain (containing LCR3 in *Figure 1A*) which plays a role in RNA-binding (*Kim et al., 2019a*), a necessary step of stress granule assembly (*Guillén-Boixet et al., 2020*; *Yang et al., 2020*). Thus, LCR relationships, which can be identified by dotplots, can be functionally important. While the importance of the LCR relationships in G3BP1 has already been studied, dotplots can be used to identify LCR relationships for any protein. By applying dotplots broadly, we can begin to ask more general questions about how patterns of LCR relationships may contribute to protein function.

While not all proteins have LCRs, some proteins almost entirely consist of LCRs and exhibit diverse LCR relationships and organization. For example, ACTB and SYTC lack LCRs, which is reflected by the lack of dense squares in their respective dotplots (*Figure 1—figure supplement 1A, B*). Other examples, such as SMN, a component of nuclear gems (*Liu and Dreyfuss, 1996*), and the nucleolar protein KNOP1 (*Grasberger and Bell, 2005*; *Larsson et al., 1999*) have more complex architectures, with multiple copies of similar LCRs, which appear with roughly equal spacing (*Figure 1—figure supplement 1C, D*). On the other hand, Nucleolin has multiple types of LCRs (*Figure 1—figure supplement 1E*), which are spatially segregated in the protein, highlighting the organizational complexity of LCRs that exists in some proteins and the ability of dotplots to make LCR relationships clear and intuitive.

As can be seen for SRRM2 and MUCIN5A, a large area in their dotplots consist of LCR signatures off the diagonal (*Figure 1B and C*), indicating that each of these proteins consist of long stretches of similar LCR sequences. For example, the dotplot of SRRM2 contains multiple regions of low complexity which cover an area corresponding to hundreds of amino acids (*Figure 1B*). SRRM2 and another LCR-containing protein SON (*Figure 1—figure supplement 1H*) were recently found to act as essential scaffolds for formation of nuclear speckles (*Sharma et al., 2010*; *Fei et al., 2017*; *Ilik et al., 2020*), suggesting that proteins which each contain long stretches of similar LCR sequences could play important roles in certain higher order assemblies. In fact, many such proteins have been found to be essential for various higher order assemblies. These include UBP2L and PRC2C (*Figure 1—figure supplement 1F, G*), which were only recently discovered to be essential for the formation of stress granules (*Youn et al., 2018*; *Sanders et al., 2020*).

Other proteins with long stretches of similar LCR sequences included mucins (MUC5A shown, *Figure 1C*), collagens and DSPP (*Figure 1—figure supplement 1I*), proteins which are essential to

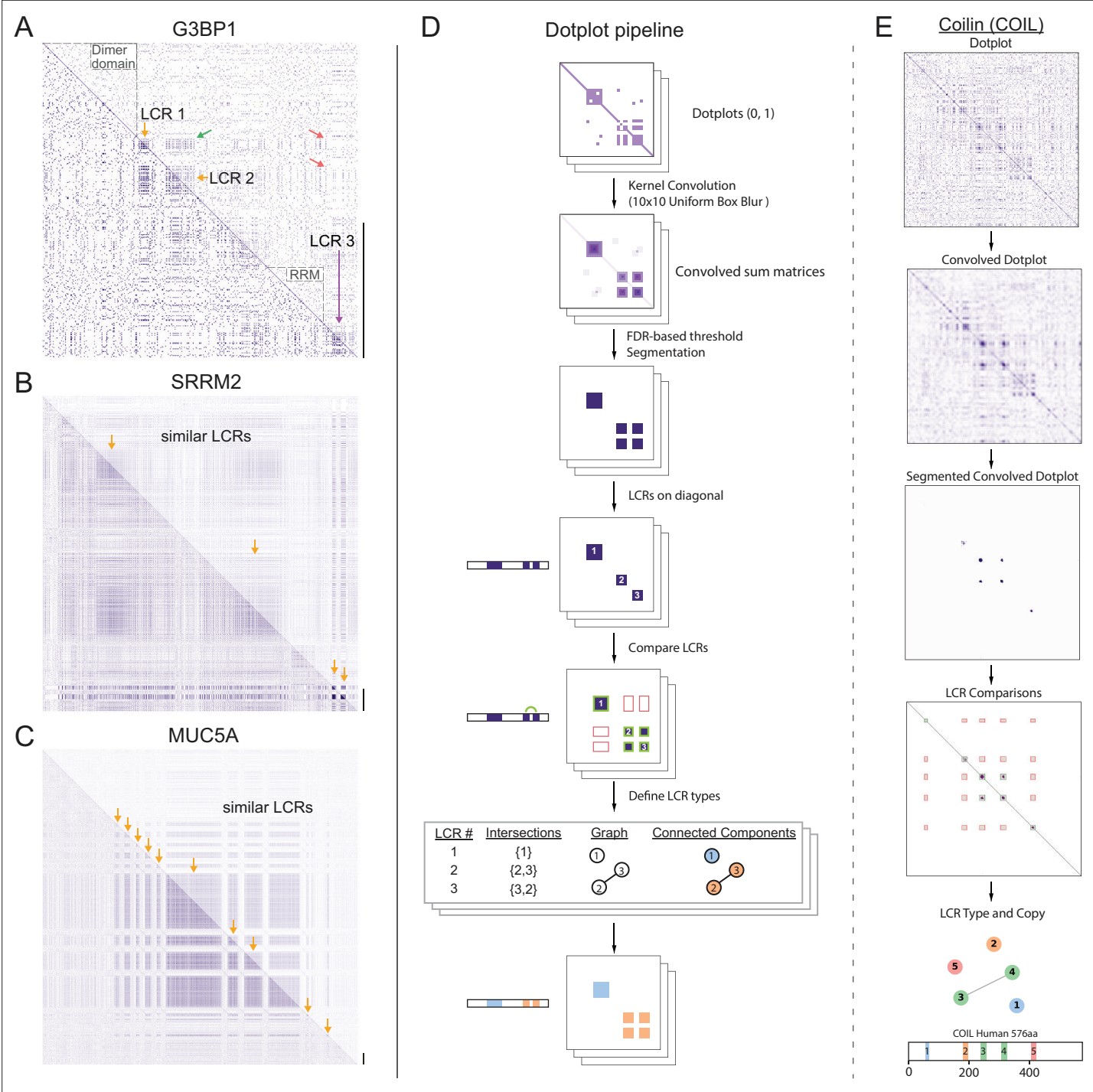

**Figure 1.** A systematic dotplot approach to reveal the relationships between low complexity regions (LCRs) in proteins. For all dotplots, the protein sequence lies from N-terminus to C-terminus from top to bottom, and left to right. Scale bars on the right of the dotplots represent 200 amino acids in protein length. (**A–C**) Raw dotplots that have not been processed with the dotplot pipeline. (**A**) Dotplot of G3BP1. Top-right half of dotplot has been manually annotated to indicate LCRs (yellow and purple arrows) and functionally important non-LC sequences (dotted lines around diagonal). Yellow arrows indicate similar LCRs. Off-diagonal regions which are informative about similar or dissimilar sequences are indicated by green arrows or red arrows, respectively. (**B**) Dotplot of SRRM2. Top-right half of dotplot has been manually annotated to indicate similar LCR sequences (yellow arrows) in SRRM2. (**C**) Dotplot of MUC5A. Top-right half of dotplot has been manually annotated to indicate similar LCR sequences (yellow arrows) in MUC5A. (**D**) Schematic of dotplot pipeline, illustrating data generation and processing. Dotplots are generated, convolved using a uniform 10x10 kernel, and segmented based on a proteome-wide FDR-based threshold (same threshold applied to all proteins in the same proteome, see Materials and methods for details). Using segmented dotplots, LCRs are identified as segments which lie along the diagonal. Pairwise off diagonal LCR comparisons

*Figure 1 continued on next page*

*Figure 1 continued*

are performed for each dotplot, and LCR relationships are represented as a graph. Connected components in this graph represent LCRs of the same type within each protein. (**E**) Sequential steps of the dotplot pipeline as performed for the human protein Coilin (COIL). Shown from top to bottom are the raw dotplot, convolved dotplot, segmented convolved dotplot, LCR-comparison plot, graph representation of LCR relationships, and schematic showing LCR position and type as called by the dotplot pipeline. Numbers represent the LCR identifier within the protein from N-terminus to C-terminus. Different colors in schematic correspond to different LCR types. See also *Figure 1—figure supplements 1–4*.

The online version of this article includes the following figure supplement(s) for figure 1:

**Figure supplement 1.** Dotplots of various human proteins.

**Figure supplement 2.** Summary statistics from systematic dotplot analysis of human proteome.

**Figure supplement 3.** Comparison of systematic dotplot analysis to existing LCR calling software, SEG and fLPS.

**Figure supplement 4.** Sequential steps of dotplot pipeline performed for several example proteins.

the formation of extracellular assemblies with a diverse variety of physical properties. Mucins are key components of mucus, a liquid/gel-like assembly of glycoproteins (reviewed in *Lai et al., 2009*), while DSPP codes for a protein which scaffolds the mineralization of teeth (*Stetler-Stevenson and Veis, 1986*; *Saito et al., 2000*; *Sreenath et al., 2003*). Although proteins which each contain such long stretches of similar LCRs, such as SRRM2, UBP2L, MUCIN5A, and DSPP, are involved in such diverse biological processes, a commonality among them is their scaffolding roles. The fact that these proteins exhibit similar LCR relationships and roles in their respective assemblies suggests that the LCR relationships revealed by dotplots can inform our understanding of protein functions.

The examples discussed above highlight the utility of dotplots in revealing LCR relationships, and raise the possibility that LCR type and copy number may play a broader role in influencing protein function. However, these features of LCRs have not been globally assessed. While several methods exist for identifying LCRs (*Wootton and Federhen, 1993*; *Promponas et al., 2000*; *Alba et al., 2002*; *Harrison, 2017*), these methods do not determine LCR relationships such as type and the respective copy numbers of these types. The ability of dotplots to both identify LCRs and provide information on LCR type and copy number presents an opportunity to develop a comprehensive and systematic tool to identify and understand these features of proteins.

## A systematic dotplot approach to identify and characterize LCRs proteome-wide

We developed a computational pipeline to extract both the positions and spatial relationships of LCRs using the 2D signature of LCRs in dotplots (*Figure 1D*, Materials and methods).

Specifically, we computationally extracted the LCRs of any protein by identifying high density regions in its dotplot through classic image processing methods, such as kernel convolution, thresholding, and segmentation (*Figure 1D*). To identify high-density regions in dotplots, we performed kernel convolution on the dotplots with a uniform 10x10 kernel, which calculates a convolved pixel intensity value from 0 to 100 based on the number of dots in that window. Regions of high density will have higher convolved pixel intensities, while regions of low density will have lower convolved pixel intensities.

In order to define LCRs in the proteome, we employed a false discovery rate (FDR)-based approach to threshold the convolved pixel intensities. For a given proteome, we generated a background model by simulating an equally sized, length-matched 'null proteome', whose sequences were generated from a uniform amino acid distribution (see Materials and methods and Appendix 1 for details). We compared the distribution of convolved pixel intensities across all proteins in the real proteome with those from the null proteome and identified the lowest convolved pixel intensity threshold which satisfied a stringent FDR of 0.002 (*Figure 1D*, *Figure 1—figure supplement 2A, B*). This threshold, which was chosen to maximize the number of called LCRs called while minimizing entropy (*Figure 1—figure supplement 2C, D*), was then applied to every protein in the human proteome to segment high-density regions in all dotplots. The segmented regions along the diagonal correspond to LCRs, while segmented regions off of the diagonal correspond to compositionally similar LCRs within the same protein (*Figure 1D*). We illustrate this process for Coilin, a scaffolding protein of Cajal bodies in the nucleus (*Figure 1E*), where dense regions in its dotplot are extracted by our systematic approach.

Across the human proteome, our approach identified 37,342 LCRs in 14,156 proteins (*Figure 1—figure supplement 2C*), with nearly 60% of LCR-containing proteins in the human proteome containing more than one LCR (*Figure 1—figure supplement 2E*). The Shannon entropy of these regions was significantly lower than that of randomly sampled sequences from the proteome, confirming that they are low complexity (*Figure 1—figure supplement 2D*). Furthermore, we observe an inverse relationship between the convolved pixel intensity threshold used for segmentation and the resulting Shannon entropy of called LCRs (*Figure 1—figure supplement 2D*). The tight relationship between these values shows that, in general, the density of points in dotplots is inversely related to the informational complexity of the corresponding sequence.

Finally, when compared to two commonly used LCR-callers, SEG (*Wootton and Federhen, 1993*) and fLPS (*Harrison, 2017*), our approach achieves a comparable performance in minimizing LCR entropy while maximizing total LCR sequence in the human proteome (*Figure 1—figure supplement 3A–D*). Furthermore, we call LCRs in regions of proteins similar to those called by other methods, as can be seen for CO1A1 and ZN579 (*Figure 1—figure supplement 3E, F*). While other approaches (*Wootton and Federhen, 1993*; *Harrison, 2017*) are more efficient at identifying the presence of LCRs, our approach allows for proteome-wide identification of LCRs while simultaneously extracting information about LCR type and copy number within proteins. Thus, by making 2D comparisons of LCRs within proteins across the proteome, our systematic dotplot approach provides additional information on the relationship between LCRs within proteins, allowing us to ask deeper questions about the role of these features in protein function.

## Comparison of LCRs defines type and copy number of LCRs across the proteome

The relationship between LCRs within LCR-containing proteins has not been studied on a proteome-wide scale, despite being important in the cases where it has been studied (*Hebert and Matera, 2000*; *Mitrea et al., 2016*; *Yang et al., 2020*). To this end, we compared all LCRs within each protein for the human proteome. The relationship between two LCRs in a protein is determined by whether or not a segmented region of the dotplot exists off the diagonal in the region corresponding to the intersection of those two LCRs. If so, we designate these two LCRs as the same 'type' (green boxes in *Figure 1D*, also see Materials and methods). The relationships between LCRs can be summarized as a graph where each LCR is a node, off-diagonal intersections between pairs of LCRs are represented as edges, and connected components are LCRs of the same type (*Figure 1D*, also see Materials and methods). This graph-based visualization is helpful for seeing complex LCR relationships within proteins (*Figure 1—figure supplement 4*), and the potential valency provided by LCRs to natural proteins.

Our approach for calling LCR type and copy number is illustrated for numerous examples with a range of different types and copy numbers (*Figure 1E*, *Figure 1—figure supplement 4*). For Coilin, we identify 4 distinct types of LCRs, with one of the types present in two copies (*Figure 1E*). Of these four types, two of them have been shown to play different roles in Coilin localization to cajal bodies (*Hebert and Matera, 2000*), showing that our systematic dotplot approach can distinguish LCR types and that these types can have different functions.

We can see from comparing the number of total and distinct LCRs across proteins in the human proteome that the range in combinations of LCRs is diverse (*Figure 2A*, *Figure 2—figure supplement 1A*), enabling different functions. Based on the number of total and distinct LCRs in a given protein, proteins can be categorized into four groups (*Figure 2B*), which each make a sizable fraction of the proteome and uniquely contribute to our understanding of how LCRs affect protein function. We will refer to these groups as 'single', 'multiple-same', 'multiple-distinct', and 'multiple-mixed' to reflect the number of total and distinct LCRs that a protein possesses.

The single LCR group, which lies in the bottom left corner (*Figure 2A*), corresponds to proteins with only a single LCR, in which we may assess the isolated function of an LCR. The multiple-same group lies along the vertical axis and corresponds to proteins with multiple LCRs, all of which are the same type. Since all of the LCRs for a given protein in this group are the same, this group is particularly useful for understanding the contribution of LCR copy number to the function of a protein. The multiple-distinct group lies along the diagonal, and corresponds to proteins with multiple LCRs, all of which are distinct from each other. This group allows for in-depth study of the relationships between

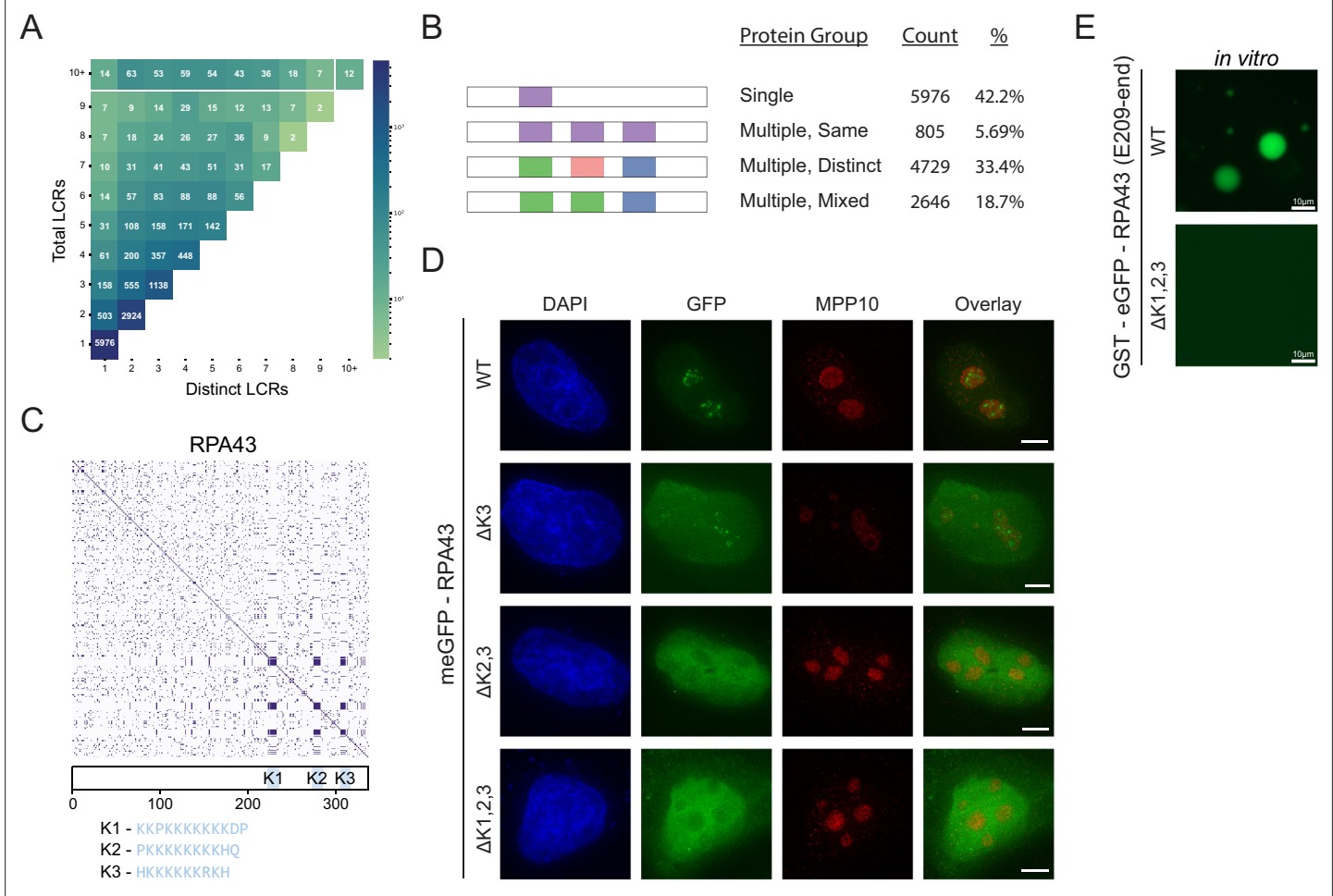

**Figure 2.** Proteome-wide definition of LCR type and copy number reveals copy number requirements for nucleolar integration of RPA43. (**A**) Distribution of total and distinct LCRs for all LCR-containing proteins in the human proteome. The number in each square is the number of proteins in the human proteome with that number of total and distinct LCRs and is represented by the colorbar. (**B**) Illustration of different protein groups defined by their LCR combinations, and the number and percentage (%) of proteins that fall into each group. Group definitions are mutually exclusive. (**C**) Dotplot and schematic of RPA43. K-rich LCRs are highlighted in blue, and are labeled K1-K3. Sequences of K1-K3 are shown below the schematic. (**D**) Immunofluorescence of HeLa cells transfected with RPA43 constructs. HeLa cells were seeded on fibronectin-coated coverslips and transfected with the indicated GFP-RPA43 constructs, and collected ~48 hr following transfection. DAPI, GFP, and MPP10 channels are shown. Scale bar is 5 μm. (**E**) Droplet formation assays using GFP-fused RPA43 C-terminus in vitro. Droplet assays were performed with 8.3 μM purified protein. Scale bar is 10 μm. See also *Figure 2—figure supplement 1*.

The online version of this article includes the following figure supplement(s) for figure 2:

**Figure supplement 1.** Supplementary information for LCR type and copy number.

different LCRs. Finally, the multiple-mixed group, which lies between the bounds of multiple-same and multiple-distinct groups, likely corresponds to more complex proteins which may be affected by both the copy number and type of LCRs they contain. By characterizing the copy number and type of LCRs across the proteome, our approach allows for proteins to be selected on the basis of these features for further study.

## LCR copy number impacts protein function

The group of proteins which have multiple LCRs of the same type presents an opportunity to specifically understand the role of LCR copy number in natural proteins. To highlight how these groups could inform us on LCR function, we sought to study the role of LCR copy number on higher order assembly by studying a protein in the 'multiple same' group.

Within the 'multiple same' group, we chose to study the RNA Polymerase I component RPA43 because it localizes to the nucleolus (*Cannon et al., 2008*), a multi-component higher order assembly. From our analysis, we found that RPA43 has three LCRs in its C-terminus which are all the same type (*Figure 2C*, *Figure 1—figure supplement 4A*). To understand the common sequences in this LCR type, we manually checked the sequences determined by our systematic analysis. All three LCRs of RPA43 contained a 10–12 amino acid block of mostly K-residues (*Figure 1—figure supplement 4A*, bottom row), which were the primary contributor to off-diagonal intersections between these LCRs and thus defined this LCR type. We chose to focus on the sequences in its three LCRs which make them the same type, the blocks of K-residues. We will refer to these K-rich blocks as K-rich LCRs of RPA43 (K1, K2, and K3 respectively).

While GFP-fused WT RPA43 localized correctly to the fibrillar center of the nucleolus, deletion of all three of its K-rich LCRs (ΔK1,2,3) led to its exclusion from the nucleolus, confirming that these LCRs are important for its higher order assembly (*Figure 2D*, *Figure 2—figure supplement 1B*). The fact that all of the LCRs of RPA43 are the same type, and that they together are required for nucleolar integration allows us to specifically study the role of LCR copy number in RPA43 higher order assembly.

We next generated all possible RPA43 mutants lacking one or more of its LCRs. Surprisingly, RPA43 mutants with two copies of its LCRs correctly localized to the nucleolus, while those containing only one of its LCRs were not (*Figure 2D*, *Figure 2—figure supplement 1C*). This result held true regardless of what combination of LCRs were present (*Figure 2—figure supplement 1C*). In fact, two of the three double mutants were strongly excluded from the nucleolus, showing that these LCRs do not uniquely contribute to RPA43 localization. Rather, it is a copy number of at least 2 of these LCRs which is required for RPA43 integration into the nucleolus.

Consistent with these results, while the recombinant GFP-fused RPA43 C-terminus phase separated into liquid droplets in vitro, the GFP-fused RPA43 C-terminus with its three K-rich LCRs specifically deleted did not (*Figure 2E*). Thus, the RPA43 C-terminus contains the sequences sufficient for higher order assembly, and the K-rich LCRs are necessary for this assembly. This result suggests that the K-rich LCRs are not merely linkers between other self-interacting elements, as deletion of such linkers tends to alter the physical properties of the assembly but not its presence (*Martin et al., 2020*). Rather, our results suggest that these K-rich LCRs participate in intermolecular interactions, and that K-rich LCR copy number may more generally contribute to protein valency. Moreover, the observation that in vivo nucleolar localization and in vitro phase separation of RPA43 require the same sequences suggest that the ability of K-rich LCRs to form higher order assemblies underlie both of these processes. Together, these results show that our approach allows for targeted experiments to understand how LCR copy number affects protein function.

## A map of LCRs

In order to understand the disparate functions of LCRs across the proteome, we wanted to understand the full breadth of LCR sequences. By using a sequence map as a foundation to integrate the features, relationships, and functions of LCRs, we could begin to relate differences in sequence to differences in LCR function. As such we took an unbiased approach to visualize the sequence space occupied by LCRs in the human proteome.

Using the LCRs identified by dotplots, we represented the amino acid composition of each LCR as a 20-dimensional vector where each dimension corresponds to the frequency of a different amino acid. Thus, each LCR will map to a point in 20-dimensional sequence space. To visualize LCR occupancy in this sequence space, we used Uniform Manifold Approximation and Projection (UMAP) (*McInnes and Healy, 2020*) to generate a two-dimensional map of all LCRs in the human proteome (*Figure 3A*, *Figure 3—figure supplement 1*).

This map shows that LCRs in the human proteome exhibit a rich diversity of sequence compositions, and do not fall into a handful of isolated groups. Generally, this LCR space has many highly occupied regions (*Figure 3A*). Using Leiden clustering (*Traag et al., 2019*), we identified 19 clusters of LCRs in this space which mostly corresponded to high frequency of an amino acid. These clusters serve as useful guides when referring to different regions of the map (*Figure 3A*). While most clusters correspond to LCRs with large contributions from a single amino acid, these clusters still have a substantial presence of many other amino acids. For example, the serine-rich cluster has regions within it that are also enriched for other amino acids in addition to serine (*Figure 3—figure supplement 2A*). These

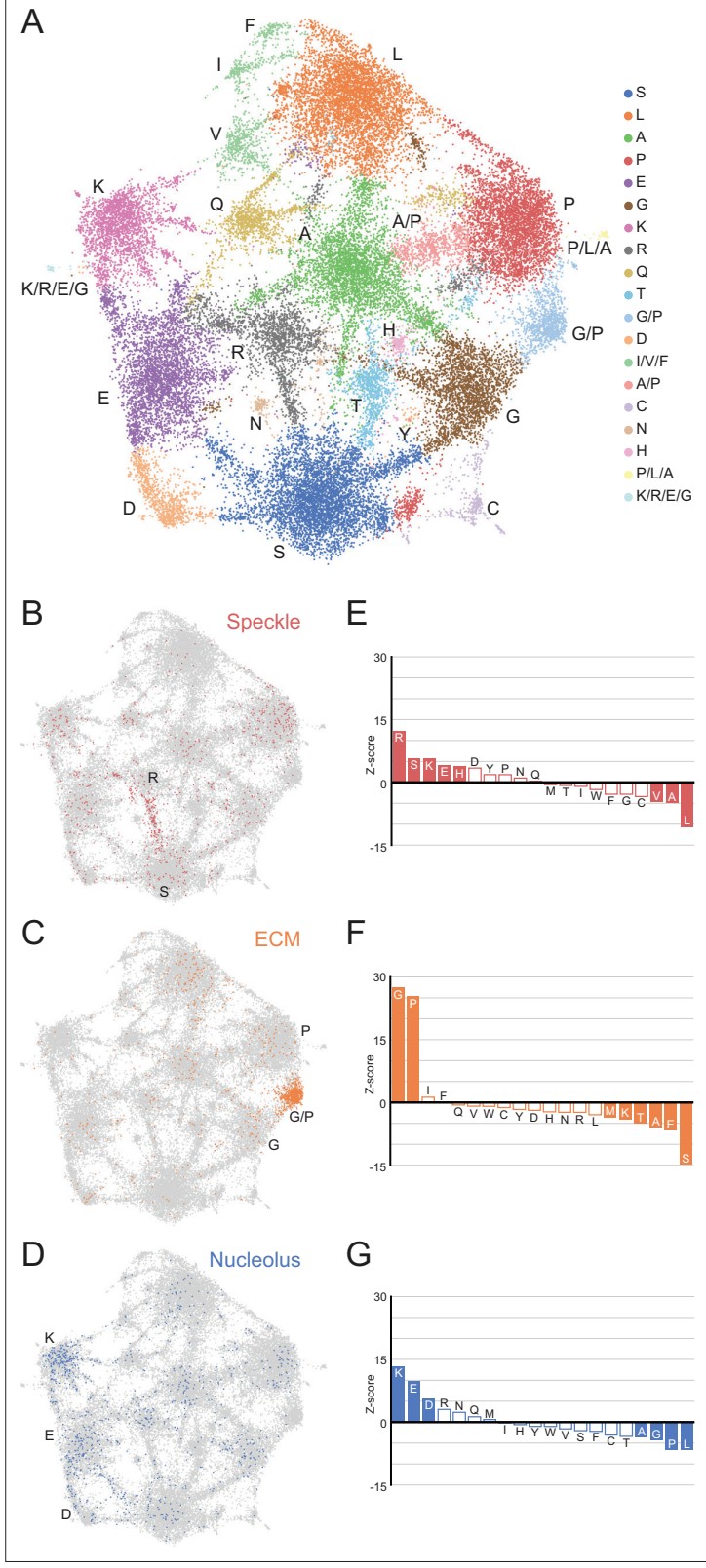

**Figure 3.** A map of LCRs captures known differences in higher order assemblies. (**A**) UMAP of all LCRs in the human proteome. Each point is a single LCR and its position is based on its amino acid composition (see Methods for details). Clusters identified by the Leiden algorithm are highlighted with different colors. Labels indicate the most prevalent amino acid(s) among LCRs in corresponding Leiden clusters. (**B**) LCRs of annotated nuclear speckle

*Figure 3 continued on next page*

*Figure 3 continued*

proteins (obtained from Uniprot, see Methods) plotted on UMAP. (**C**) Same as (**B**), but for extracellular matrix (ECM) proteins. (**D**) Same as (**B**), but for nucleolar proteins. (**E**) Barplot of Wilcoxon rank sum tests for amino acid frequencies of LCRs of annotated nuclear speckle proteins compared to all other LCRs in the human proteome. Filled bars represent amino acids with Benjamini-Hochberg adjusted p-value < 0.001. Positive Z-scores correspond to amino acids enriched in LCRs of nuclear speckle proteins, while negative Z-scores correspond to amino acids depleted in LCRs of nuclear speckle proteins. (**F**) Same as (**E**), but for extracellular matrix (ECM) proteins. (**G**) Same as (**E**), but for nucleolar proteins. See also *Figure 3—figure supplements 1–3*.

The online version of this article includes the following figure supplement(s) for figure 3:

**Figure supplement 1.** Amino acid frequency distributions on human proteome UMAP from *Figure 3A*.

**Figure supplement 2.** Nuanced sequence differences among LCRs correspond to their positions in the UMAP.

**Figure supplement 3.** LCRs of known higher order assemblies annotated on onto human proteome UMAP from *Figure 3A*.

regions of the S-rich cluster are typically closer to the main cluster corresponding to the other amino acid, highlighting the richness in diversity of LCR compositions, even within one single cluster.

Strikingly, many clusters are 'connected' to other clusters through bridge-like connections, which are much more prominent between certain clusters (*Figure 3A*, *Figure 3—figure supplement 2*). This indicates that some combinations of amino acids commonly co-occur to varying degrees within LCRs which occupy these bridges, while other combinations of amino acids do not co-occur as often. While cluster definitions are discrete, the amino acid compositions of the LCRs that lie along these bridges are continuous (*Figure 3—figure supplement 2B, C*). In some cases, such as in the G/P-rich cluster between the main G- and P-rich clusters, these bridges are large enough to form their own clusters (*Figure 3A*, *Figure 3—figure supplement 2B*). The observation that LCRs exhibit a gradual, continuous shift in LCR composition from one end of the bridge to the other raises the possibility that any properties sensitive to the composition of these LCRs may exhibit a similarly gradual and continuous variation, increasing the potential complexity of interactions formed by LCRs.

This map reveals the high degree of nuanced sequence variation that exists in natural LCRs and that certain amino acids coexist to varying degrees in LCRs. By capturing the variation in all LCRs, this global map provides an intuitive foundation for understanding how biological and physical properties of LCRs relate to their sequence.

## Higher order assemblies map to specific regions in LCR sequence space

LCRs of certain compositions play important roles in specific higher order assemblies. To gain insight into what functions are represented in different regions of LCR sequence space, we decided to see if higher order assemblies preferentially occupy certain regions in the map.

To do this, we mapped annotations of known higher order assemblies to the LCR map. Nuclear speckle proteins, which are commonly localized by LCRs known as RS-domains (*Cáceres et al., 1997*; *Boucher et al., 2001*), primarily populated a bridge between the R and S clusters in LCR sequence space (*Figure 3B*), and were significantly enriched in both of these amino acids (*Figure 3E*). LCRs of extracellular matrix (ECM) proteins were heavily concentrated in a G/P-rich region (*Figure 3C and F*), reflecting the many, long collagen proteins in humans. LCRs of nucleolar proteins largely mapped to the K-rich and E-rich clusters in LCR sequence space (*Figure 3D and G*), consistent with nucleolar localization signals possessing K-rich sequences (*Scott et al., 2010*). Other higher order assemblies also had biased occupancy of specific regions in the LCR sequence space, including the centrosome and nuclear pore complex (*Figure 3—figure supplement 3A, B*). Wilcoxon rank-sum tests for each of the 20 amino acids confirmed that these spatial biases in the LCR map corresponded to actual differences in LCR composition, independent of the map (*Figure 3E–G*, *Figure 3—figure supplement 3E, F*). Conversely, some higher order assemblies are known to not have many individual proteins which share a specific LCR composition, including stress granules for which RNA is a major contributor (*Guillén-Boixet et al., 2020*; *Sanders et al., 2020*), and PML bodies which depend on SUMOylation of non-LC sequences (*Shen et al., 2006*). As expected for these cases, there was neither a spatial bias in the LCR map, nor statistically significant enriched amino acids (*Figure 3—figure supplement 3C, D, G, H*).

The ability of the map to highlight the biased LCR compositions of certain higher order assemblies demonstrates that we can capture how differences in sequence correspond to known differences in function. Thus, the LCR map allows us to interrogate the relationship between less understood regions of LCR space and protein function.

## A unified view of LCRs reveals scaffold-client architecture of E-rich LCR-containing proteins in the nucleolus

When examining the distribution of nucleolar protein LCRs across the LCR map, we found that in addition to the K-rich cluster, the E-rich cluster was significantly occupied (*Figure 3D and G*). Nucleolar LCRs were significantly more likely than LCRs of speckle proteins to have a high frequency of E residues, but not D residues (*Figure 4—figure supplement 1A*). This observation suggested that E-rich LCRs may play a nucleolus-specific role.

To see if the nucleolar E-rich LCRs had any features which could give insight into their role in the nucleolus, we first integrated our dataset with biophysical predictions relevant to higher order assemblies (*Figure 4—figure supplement 2*). We analyzed LCRs in the top 25th percentile of K or E frequency in the nucleolus, which we refer to as K-enriched and E-enriched respectively (*Figure 4A*). The intrinsic disorder scores (IUPred2) among these K- and E-enriched nucleolar LCRs are similar (*Figure 4B*). However, while K-enriched nucleolar LCRs exhibited a unimodal distribution of ANCHOR scores (the probability of a disordered sequence to become ordered upon binding to a globular protein partner), E-enriched LCRs exhibited a bimodal distribution where one peak had much higher ANCHOR scores (*Figure 4C*). This raises the possibility that they fulfill non-overlapping roles in the structure of the nucleolus.

To gain a better understanding of the contribution of E-rich LCRs to the nucleolus, we looked at the type and copy number of these LCRs among nucleolar proteins with E-enriched LCRs. Of the 319 LCR-containing nucleolar proteins, 137 had at least one E-enriched LCR. Moreover, the distribution of total vs distinct LCRs of nucleolar proteins containing E-enriched LCRs showed that many of these proteins were of the multiple-mixed type, with some even reaching 22 total LCRs across four distinct LCR types (*Figure 4D*). From this analysis, we hypothesized that the nucleolar proteins with many E-rich LCRs act as scaffolds, while those with few E-rich LCRs act as clients.

Among proteins with E-enriched LCRs, we chose TCOF as a candidate scaffold because it contained the most E-enriched LCRs (*Figure 4E*) and was a multiple-mixed protein high in total LCRs but low in distinct LCRs (*Figure 4D*). Two of these LCRs were K-rich LCRs in the C-terminus of TCOF, similar to the homopolymeric stretches we observed in RPA43. Moreover, TCOF has a striking pattern of several evenly spaced E-rich LCRs which make up 15/22 of its LCRs, as illustrated by its dotplot and UMAP (*Figure 1—figure supplement 4D*, *Figure 4—figure supplement 1B*). TCOF normally localizes to the nucleolus, as indicated by the nucleolar marker Fibrillarin and Hoechst-negative regions surrounded by heterochromatin (upper row in *Figure 4F*, *Figure 4—figure supplement 1D*).

To test if TCOF is a scaffold, we wanted to assess its ability to assemble outside of the nucleolus. To attempt this, we deleted the K-rich sequences in the C-terminus of TCOF (TCOF ΔK, *Figure 4—figure supplement 1C*). If TCOF is indeed a scaffold and this scaffolding property is mediated by its E-rich LCRs, then this region of TCOF may be sufficient for assembly in the nucleoplasm. When expressed in cells, TCOF ΔK forms assemblies similar in size and shape to WT TCOF (*Figure 4F*, green channel). However, when assessed both by Fibrillarin costaining and heterochromatin-surrounded Hoechst-negative regions, TCOF ΔK was able to form assemblies outside of the nucleolus (*Figure 4F*, *Figure 4—figure supplement 1D*), showing that the region of TCOF containing its E-rich LCRs is sufficient for assembly.

The observation that TCOF ΔK is sufficient to form assemblies allowed us to further test if nucleolar proteins with a low number of E-enriched LCRs are recruited to these assemblies as clients. We chose to assess UBF1 and RPA1, which from our analysis have 1 and 2 E-enriched LCRs (*Figure 4E*), respectively, and are both key nucleolar proteins. RPA1 (also known as POLR1A) is a core subunit of RNA polymerase I, which synthesizes rRNA, and UBF1 is a key transcription factor facilitating Pol I transcription initiation (*Bell et al., 1988*). Both RPA1 and UBF1 colocalize with WT TCOF in the nucleolus (upper row in *Figure 4G*, *Figure 4—figure supplement 1E*). When we assessed the localization of RPA1 and UBF1 in cells expressing TCOF ΔK, we observed that both of these proteins now colocalized with TCOF ΔK. Moreover, the enrichment of RPA1, which contains two E-rich LCRs, in TCOF

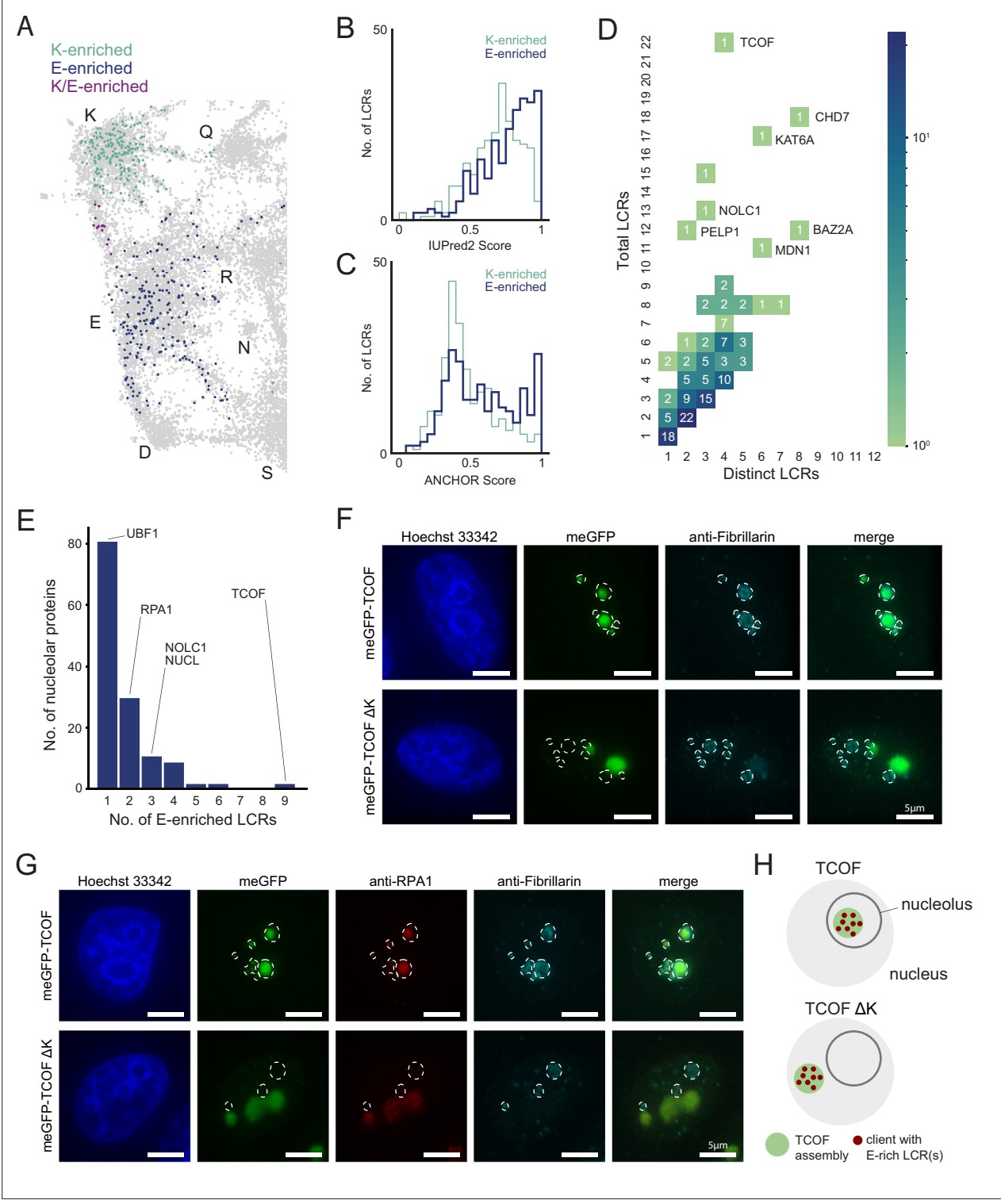

**Figure 4.** An integrated LCR map reveals scaffold-client architecture of E-rich LCR-containing proteins in the nucleolus. (**A**) Nucleolar LCRs which are E-enriched (top 25% of nucleolar LCRs by E frequency), K-enriched (top 25% of nucleolar LCRs by K frequency), or K/E-enriched (both E- and K-enriched) plotted on close-up of K/E-rich regions of UMAP from *Figure 3A*. (**B**) Distribution of IUPred2 scores for K-enriched and E-enriched nucleolar LCRs. (**C**) Distribution of ANCHOR scores for K-enriched and E-enriched nucleolar LCRs. (**D**) Distribution of total and distinct LCRs for all nucleolar LCR-containing proteins in the human proteome with at least one E-enriched LCR. The number in each square is the number of proteins with that number of total and distinct LCRs and is represented by the colorbar. Several proteins with many LCRs are labeled directly to the right of their coordinates on the graph. (**E**) Distribution of the number of E-enriched LCRs for nucleolar proteins. Proteins with zero E-enriched LCRs are not included.

*Figure 4 continued on next page*

*Figure 4 continued*

(**F**) Immunofluorescence of cells transfected with meGFP-TCOF or meGFP-TCOF ΔK, stained with anti-fibrillarin antibody, and Hoechst 33342 (see Materials and methods). Merge image is an overlay of the meGFP and Fibrillarin images. Dotted line represents the outline of fibrillarin-positive regions, marking the nucleolus. Scale bars are 5 µm. (**G**) Immunofluorescence of cells transfected with meGFP-TCOF or meGFP-TCOF ΔK, stained with anti-fibrillarin antibody, anti-RPA1 antibody, and Hoechst 33342 (see Materials and methods). Merge image is an overlay of the meGFP, RPA1, and Fibrillarin images. Dotted line represents the outline of fibrillarin-positive regions, marking the nucleolus. Scale bars are 5 µm. (**H**) Illustrated schematic of client recruitment by TCOF or TCOF ΔK. See also *Figure 4—figure supplements 1 and 2*.

The online version of this article includes the following figure supplement(s) for figure 4:

**Figure supplement 1.** Supplementary Data for nucleolar E-rich LCRs and TCOF.

**Figure supplement 2.** Biophysical predictions of LCRs mapped onto human proteome UMAP from *Figure 3A*.

ΔK assemblies appears greater than enrichment of UBF1, which contains one E-rich LCR (*Figure 4G*, *Figure 4—figure supplement 1E*). These results show that a protein with many E-rich LCRs (TCOF) is a scaffold, and that proteins with few E-rich LCRs act as clients (*Figure 4H*).

More broadly, these results suggest that there are signatures of scaffold and client proteins of higher order assemblies in their LCR copy numbers, and that by looking for these signatures, we can discover previously unappreciated scaffolds and their respective clients. Thus, a unified view of LCRs allows us to uncover the scaffold-client architecture of a higher order assembly and provides a framework for understanding of the role of LCRs in other regions of sequence space.

## An expanded map of LCRs across species

The true breadth of LCR functions is not captured in any single species. The relationship between regions of LCR space and higher order assemblies raises several questions about whether a unified view of LCRs can more generally relate the functions of LCRs across species. In species where the existence of a given assembly such as the nucleolus is conserved, is occupancy of the sequence space also conserved? Similarly, does emergence of a certain higher order assembly across evolution such as the extracellular matrix correlate with the occupancy of a certain region of sequence space? Conversely, many species have distinct higher order assemblies with different functions and physical properties from those in humans, such as plants and fungal cell walls. Do these assemblies occupy a distinct region of sequence space, or do they use a sequence space that is occupied in humans?

To answer these questions, we wanted to capture the entire breadth of LCR sequence space across species, so that we could concurrently compare how different species occupy this sequence space. We applied our dotplot and dimensionality reduction approach to the proteomes of *E. coli*, *S. cerevisiae*, *A. thaliana*, *C. elegans*, *D. melanogaster*, *D. rerio*, *M. musculus*, and *H. sapiens*. This allowed us to simultaneously compare between prokaryotes and eukaryotes, among fungi, plants, and animals, and across metazoans.

After we confirmed that we indeed call LCRs in all of these species (*Figure 5—figure supplement 1*), we generated a map of the full breadth of LCR sequence space across these species (*Figure 5A*, *Figure 5—figure supplement 2*).

This map gave us a general view of the distribution and properties of LCRs of different species (*Figure 5—figure supplements 3 and 5*). Some regions of sequence space were occupied to some degree by all species analyzed, while others appeared specific to certain species (*Figure 5—figure supplements 2–4*). Furthermore, certain bridges were observed in several species, while a few bridges were predominantly occupied in specific species (*Figure 5—figure supplements 3 and 4C*), indicative of the inter-species diversity of LCR sequences.

## Conserved and diverged higher order assemblies are captured in LCR sequence space

With our expanded map, we wanted to see if higher order assemblies which were conserved or diverged between species corresponded to similarities and differences in the occupancy of LCR space. We mapped nucleolar annotations from *S. cerevisiae* and *H. sapiens* to compare the occupancy of nucleolar LCRs in these species. The space occupied by nucleolar LCRs from yeast and human were both common to the K-rich cluster as well as the E/D-rich clusters (*Figure 5B*, *Figure 5—figure supplement 6A, B*), suggesting that the compositions of LCRs participating in the nucleolus are conserved

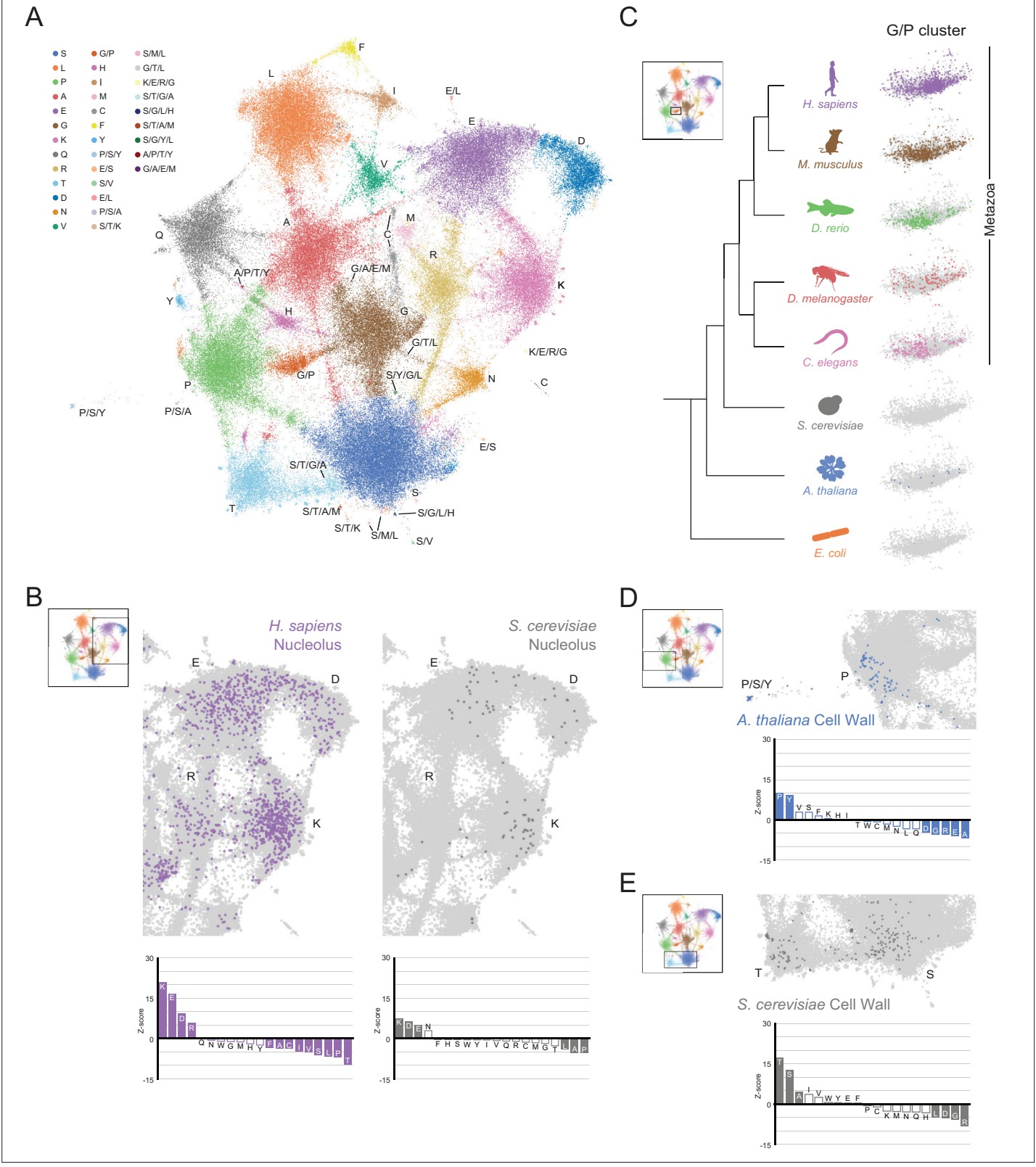

**Figure 5.** The conservation and emergence of higher order assemblies is captured in an expanded LCR map across species. (**A**) UMAP of LCR compositions for all LCRs in the human (*H. Sapiens*), mouse (*M. musculus*), zebrafish (*D. rerio*), fruit fly (*D. melanogaster*), worm (*C. elegans*), Baker's yeast (*S. cerevisiae*), *A. thaliana*, and *E. coli* proteomes. Each point is a single LCR and its position is based on its amino acid composition (see Materials and methods for details). Leiden clusters are highlighted with different colors. Labels indicate the most prevalent amino acid(s) among LCRs

*Figure 5 continued on next page*

*Figure 5 continued*

in corresponding leiden clusters. (**B**) Close-up view of UMAP in (**A**) with LCRs of human nucleolar (left) and yeast nucleolar (right) proteins indicated. (Bottom) Barplot of Wilcoxon rank sum tests for amino acid frequencies of LCRs of annotated human nucleolar proteins (left) and yeast nucleolar proteins (right) compared to all other LCRs in the UMAP (among all included species). Filled bars represent amino acids with Benjamini-Hochberg adjusted p-value < 0.001. (**C**) Close-up view of G/P-rich cluster from UMAP in (**A**) across species as indicated. LCRs within the G/P-rich cluster from each species are colored by their respective species. Species are organized by their relative phylogenetic positions. (**D**) Close-up view of UMAP in (**A**) with LCRs of *A. thaliana* cell wall proteins indicated. Barplot of Wilcoxon rank sum tests for amino acid frequencies of LCRs of annotated *A. thaliana* cell wall proteins compared to all other LCRs in the UMAP (among all included species). Filled bars represent amino acids with Benjamini-Hochberg adjusted p-value < 0.001. (**E**) Same as (**D**) but with LCRs of *S. cerevisiae* cell wall proteins. See also *Figure 5—figure supplements 1–6*.

The online version of this article includes the following figure supplement(s) for figure 5:

**Figure supplement 1.** Summary statistics from systematic dotplot analysis across species.

**Figure supplement 2.** Amino acid frequency distributions mapped onto expanded UMAP from *Figure 5A*.

**Figure supplement 3.** LCRs of individual species mapped onto expanded UMAP from *Figure 5A*.

**Figure supplement 4.** Examples of species-specific clusters in the expanded UMAP from *Figure 5A*.

**Figure supplement 5.** Biophysical predictions of LCRs mapped onto the expanded UMAP from *Figure 5A*.

**Figure supplement 6.** Higher order assemblies in different species annotated on the expanded UMAP from *Figure 5A*.

across a large evolutionary distance, including the E-rich sequences discussed above (*Figure 4*). Similarly, when comparing between speckle annotations for *A. thaliana* and *H. sapiens*, we found that the R/S-rich bridge between the R-rich and S-rich clusters was occupied for each (*Figure 5—figure supplement 6C-F*). In areas in which higher order assemblies are conserved between species, occupancy of LCR sequence space is generally conserved.

Furthermore, changes in the LCR sequence space corresponded to differences in higher order assemblies, such as the extracellular matrix which occupied the G/P cluster in humans. While *E. coli*, *S. cerevisiae*, and *A. thaliana* had nearly no LCRs in the G/P cluster, this cluster was much more occupied in metazoans (*Figure 5C*), corresponding with the emergence of collagens, a hallmark of the metazoan lineage (*Hynes, 2012*). This difference in G/P occupancy could not be explained by differences in the total number of LCRs in these species, since *A. thaliana* had more LCRs than *C. elegans* but much lower occupancy in the G/P cluster. Within metazoans, although this cluster was occupied in *C. elegans* and *D. melanogaster*, it was more heavily occupied in vertebrates. Many LCRs existed in this cluster in *D. rerio*, and even more in *M. musculus* and *H. sapiens*, which spanned most of the space in this G/P cluster. Again, the difference in G/P cluster occupancy could not be explained by the total number of LCRs in each species, as *D. melanogaster* had more LCRs than all of the vertebrates, but lower G/P occupancy (*Figure 5C*, *Figure 5—figure supplement 3*). The gradual differences in occupancy of the G/P cluster between the metazoan species (*Figure 5C*) correlated with the expansion of the extracellular matrix across metazoans (reviewed in *Hynes, 2012*), highlighting that the LCR map traces the progression of a higher order assembly across evolution.

Across longer evolutionary distances, different species-specific higher order assemblies mapped to unique regions of LCR sequence space. LCRs of cell wall proteins of *A. thaliana*, for example, primarily mapped to the P-rich cluster and a nearby P/S/Y-rich cluster (*Figure 5D*, *Figure 5—figure supplement 6G*), reflecting the set of hydroxyproline-rich cell wall proteins which include extensins, arabinogalactan proteins (AGPs) and proline-rich proteins (PRPs). Extensins, which have SPPPP motifs, are known to be important scaffolds for the assembly of the cell wall, in which they are thought to form self-assembling networks to organize pectin (*Cannon et al., 2008*; *Sede et al., 2018*). In *S. cerevisiae*, LCRs of cell wall proteins mapped to the S-rich and T-rich clusters (*Figure 5E*, *Figure 5—figure supplement 6H*), which included flocculation proteins. These S- and T-rich LCRs are often sites for O-mannosylation in mannoproteins, which is crucial for the integrity of the cell wall (*Gentzsch and Tanner, 1996*; *González et al., 2012*; *Neubert et al., 2016*).

Our approach allowed us to answer several general questions about the relationships between the sequence space occupied by LCRs and their functions. Firstly, we show that when a given assembly is conserved, occupancy of the corresponding LCR sequence space is also conserved. Secondly, the emergence of a higher order assembly can correspond to the population of a previously unoccupied sequence space. Finally, higher order assemblies with different physical properties occupy different regions of sequence space, even when they fulfill similar roles in their respective species. While these

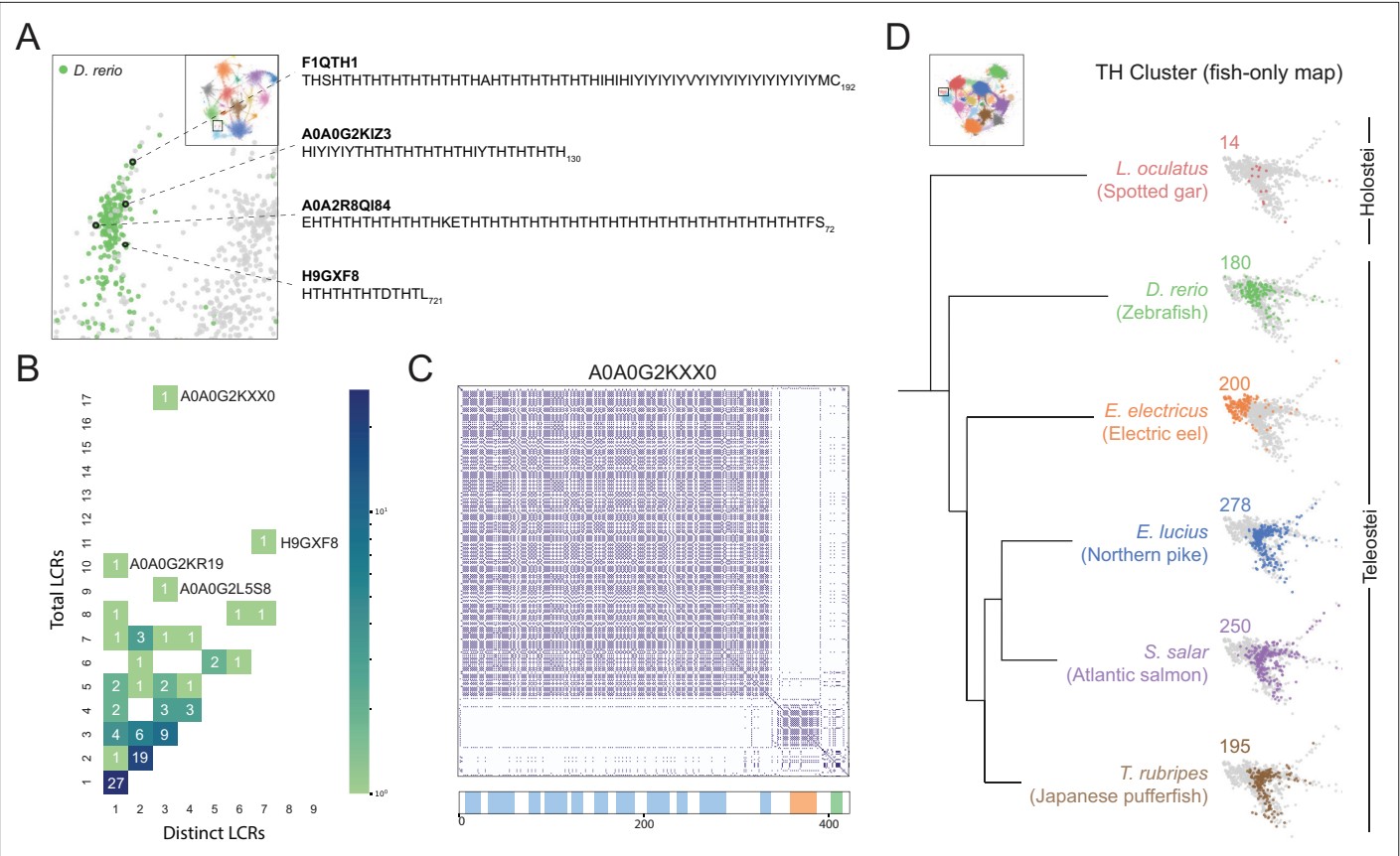

**Figure 6.** A conserved, teleost-specific T/H rich cluster exhibits signatures of higher order assemblies. (**A**) Close up of T/H-rich region in UMAP shown in *Figure 5A*. LCRs of *D. rerio* are indicated in green, LCRs of all other species in UMAP are indicated in grey. Specific LCRs are circled and the dotted lines point to their parent protein and sequences (right). For all LCRs shown, the subscript at the end of the sequence corresponds to the ending position of the LCR in the sequence of its parent protein. (**B**) Distribution of total and distinct LCRs for all *D. rerio* proteins with at least one LCR in the T/H-rich region. The number in each square is the number of proteins with that number of total and distinct LCRs and is represented by the colorbar. Several proteins with many LCRs are labeled directly to the right of their coordinates on the graph. (**C**) Dotplot of A0A0G2KXX0, the *D. rerio* protein in the T/H-rich region with the largest number of total LCRs. Schematic showing positions of LCRs called from dotplot pipeline are shown below. Different colors in schematic correspond to different LCR types within A0A0G2KXX0. (**D**) T/H-rich cluster in UMAP generated from LCRs in proteomes of zebrafish (*D. rerio*), Spotted gar (*L. oculatus*), Electric eel (*E. electricus*), Northern pike (*E. lucius*), Atlantic salmon (*S. salar*), and Japanese pufferfish (*T. rubripes*). LCRs within the T/H-rich cluster from each species are colored by their respective species. The number above each UMAP cluster is the number of LCRs from each species inside that cluster. Species are organized by their relative phylogenetic positions and members of Teleostei and Holostei are indicated. See also *Figure 6—figure supplement 1*.

The online version of this article includes the following figure supplement(s) for figure 6:

**Figure supplement 1.** Number and proportion of T/H-rich LCRs across fish species.

## A teleost-specific T/H cluster contains scaffold-like proteins and evolutionary signatures of higher order assemblies

Beyond the spaces where known higher order assemblies exist, there is a vast region of LCR sequence space which is unexplored. The observation that known higher order assemblies only occupy a subset of this space raises the question of whether the other, unexplored regions of sequence space may harbor LCRs of previously unknown higher order assemblies. To this end, we searched for signatures of higher order assemblies in previously undescribed regions of sequence space which are differentially occupied across species.

Upon comparing sequence space occupancy between species, we found various species-specific regions which lacked detailed annotations, one of which was the T/H-rich cluster specific to *D. rerio*

(*Figure 6A*, *Figure 5—figure supplement 3*). Many of the LCRs in this cluster included direct TH repeats (*Figure 6A*), which was of particular interest because these amino acid residues may facilitate assembly via several mechanisms, including polar interactions through threonine or cation-pi interactions of histidine. In addition to these mechanisms, they may also acquire mixed-charge properties under certain phosphorylation/pH conditions and behave like other LCRs composed of mixed charges, which are known to form higher order assemblies (*Greig et al., 2020*). Therefore, we decided to further investigate this T/H-rich cluster, which contained 97 proteins with T/H-rich LCRs. To see if there could be signatures of higher order assemblies in this cluster, we looked for proteins which may be more client or scaffold-like in terms of their LCR relationships, as we found for nucleolar E-rich LCR-containing proteins. This analysis showed that proteins with T/H-rich LCRs have a wide distribution of total and distinct LCRs (*Figure 6B*). Of particular interest were proteins with a high number of T/H LCRs, which could be similar to scaffold proteins like TCOF or SRRM2. For example, in the plot of total vs. distinct LCRs for the T/H-rich cluster, protein A0A0G2KXX0 had 17 total LCRs and only three distinct LCR types. Of these, 15 were T/H-rich LCRs, with only 1 LCR in each of the other distinct types (*Figure 6C*). Thus, T/H LCRs and all of the properties which come with a T/H composition, exist in high copy number in some proteins, suggesting that these proteins may scaffold a higher order assembly.

Next, we wanted to determine if the T/H-rich LCRs in zebrafish might have evolutionary signatures of higher order assemblies which may hint at whether they are functionally important. Given that conserved higher order assemblies tend to correspond to conserved occupancy in regions of sequence space, we tested if occupancy of this T/H sequence space was conserved in fishes. We used our dotplot and UMAP approach to identify and cluster LCRs from a range of fishes from the clade Actinopterygii (*Hughes et al., 2018*), to which zebrafish belongs. The six species we analyzed were zebrafish, electric eel, northern pike, Atlantic salmon, Japanese pufferfish, and spotted gar (*D. rerio*, *E. electricus*, *E. luciu*s, *S. salar*, *T. rubripes*, and *L. oculatus*, respectively). Of these fishes, all but the spotted gar substantially occupied the T/H-rich cluster. The fishes which heavily occupied the T/H-rich cluster each contained ~200 T/H LCRs, while the spotted gar was only lightly occupied with 14 LCRs (*Figure 6D*, *Figure 6—figure supplement 1A*). This difference could not be attributed to differences in total LCR count among these species (*Figure 6—figure supplement 1A, B*). However, this difference in occupancy of the T/H cluster correlated exactly with evolutionary relationships between these fishes. Those containing T/H-rich LCRs belonged to Teleostei, while the spotted gar, which did not, belonged to Holostei, a group which diverged from Teleostei in Actinopterygii. Moreover, the seven species we analyzed outside of Actinopterygii had a very low number of T/H-rich LCRs as well (*Figure 6A*, *Figure 5—figure supplement 3*), which strongly suggests that T/H-rich LCRs are teleost specific and may form a conserved higher order assembly in these species.

Altogether, our approach was able to unearth conserved LCR compositions, with scaffold-like distributions within their parent proteins. These results not only demonstrate the existence of unexplored LCRs with signatures of higher order assemblies, but also that our understanding of LCR sequences and their corresponding functions in disparate assemblies may be connected by this unified view of LCRs.

## Discussion

Here, we have established a systematic approach to study LCRs, providing a unified view of how the sequences, features, relationships and functions of LCRs relate to each other. This unified view enabled us to gain insight into the role of LCRs in multivalent interactions, higher order assemblies, and organismal structures. Moreover, this framework for understanding LCRs begins to answer fundamental questions about how LCRs encode their functions.

### How can low complexity sequences capture the diversity of LCR function?

While the functions of proteins are encoded in their sequence, it has been difficult to assign functions to LCRs. Any mapping between LCR function and sequence space presents a question of how the many disparate functions of LCRs can exist in a space which only employs a few amino acids at a time.

In our LCR map, we find that natural LCRs distribute across a continuum of sequence space. Such nuanced differences in amino acid composition might enable similarly nuanced differences in the

functions they encode. One known example of such nuanced LCR function is in the acidic LCR of G3BP1, which interacts with and inhibits its RNA-binding RGG LCR (*Guillén-Boixet et al., 2020*; *Yang et al., 2020*). This inhibitory activity of the acidic LCR is independent of the primary sequence of the acidic LCR, and is abolished by substitution of negatively charged glutamic acid for neutral glutamine residues (*Yang et al., 2020*). These results suggest that gradual changes in the ratio of glutamine to glutamic acid may alter the inhibitory activity of such an LCR. Given that we observe a bridge connecting the E and Q clusters, such a range in activity may exist across proteins in the human proteome. We observe various other bridges, highlighting that meaningful functional differences may exist in the nuanced compositional differences of naturally occurring LCRs.

Differences in amino acid composition also imply differences in sequence. It follows that functional consequences downstream of sequence, such as post-translational modifications, can be affected by differences in composition. We have shown that several bridge-like connections exist between the clusters for serine and other amino acids in the LCR map. One well understood kinase, CK2, binds and phosphorylates serines in acidic contexts (*Rusin et al., 2017*). Interestingly, bridge-like connections exist between both S and D, and S and E in the LCR map, raising the possibility that their physical properties can be regulated to different extents by CK2. Notably, TCOF, which has many E-rich LCRs in the bridge between S and E, has been shown to be regulated by CK2 (*Werner et al., 2015*; *Werner et al., 2018*). Furthermore, our data suggests that TCOF is a scaffolding protein for other proteins with E-rich LCRs. Thus, the ability of TCOF to scaffold and recruit its key nucleolar clients may be modulated by phosphorylation of its LCRs. Differences in post-translational modifications may represent an additional layer by which LCRs can encode biological functions. More broadly, the functional consequences of nuanced differences in other regions of LCR space can now be systematically studied.

## Implications of bridges between certain amino acids in LCR space

Looking more generally at the LCR maps, the presence or absence of certain bridges connecting clusters may correspond to informative relationships between pairs of amino acids. We found that various bridges exist in the map, including the bridges between L and each of I, F, and V, the K - E - D axis, and the G/P and R/S bridges.

Some of these bridges represent mixtures of similar residue properties, such as hydrophobic or negatively-charged amino acids. These findings are consistent with the hypothesis that some sets of amino acids with similar physical properties may be redundant, and thus varying combinations of them are not selected against. Interestingly, while R and K are both positively charged, basic residues, the region between these clusters was poorly populated, suggesting that these residues may not always be interchangeable in LCRs. This is consistent with known differences between R and K, such as the ability of R to participate in stacking interactions. In fact, recent evidence showed that the physical properties of R and K substantially differ, while the difference between D and E is much more subtle (*Wang et al., 2018*; *Greig et al., 2020*; *Fossat et al., 2021*). Thus, while co-occurrence of similar amino acids may not be entirely surprising, a lack of co-occurrence between seemingly similar amino acids may point towards interesting differences between them.

Likewise, while dissimilar amino acids may not often co-occur in LCRs, the presence of bridges with dissimilar amino acids may represent combinations which have emergent functions. It has not escaped our notice that R/S and G/P are combinations of dissimilar amino acids which correspond to functional, conserved higher order assemblies–the speckle and extracellular matrix. In these cases, it is known how the combinations of amino acids may enable emergent properties, such as mixed charge domains (*Greig et al., 2020*) or tight-packing polyproline helices (*Cowan and McGAVIN, 1955*; *Ramachandran and Kartha, 1955*; *Rich and Crick, 1955*). However, certain combinations exist in which the properties are not well understood, such as N/S or T/H, or only beginning to be explored such as H/Q (*Gutierrez et al., 2022*). Thus, we hypothesize that the existence of bridges between dissimilar amino acids may correspond to LCRs with specific emergent properties. These types of LCRs represent open, unexplored regions of LCR space for which the relationship between sequence and function has yet to be determined.

## A unified LCR map relates disparate higher order assemblies across species

The ability of the LCR map to capture certain higher order assemblies raises questions of what the LCRs in other parts of the map may tell us about their functions. While we do not interpret that all

LCRs must be involved in higher order assembly, the observation that LCRs of different higher order assemblies populated different regions of the same sequence space allows us to consider if there are similarities among the roles of LCRs which give us insight into LCR function. For example, the nucleolus is a liquid assembly of protein, RNA, and DNA essential for ribosome biogenesis, while the extracellular matrix is a solid/gel-like assembly of glycoproteins scaffolded by long collagen fibers. The K-rich LCRs of nucleolar proteins such as RPA43 are required for their higher order assembly and integration into the nucleolus, while the G-P-P motif-containing LCRs in various collagens form key assemblies in the ECM (*Timpl et al., 1981*; *Mould and Hulmes, 1987*; *Hansen and Bruckner, 2003*) and the G/V/P-rich LCRs of elastin assemble to provide ECM elasticity (*Urry et al., 1974*, *Rauscher et al., 2006*). Although these examples are vastly different in physical properties, a common theme is that the LCRs enable the integration and assembly of various biomolecules in biological structures.

If this is the case, we may gain insight into the structures and organizations of species by comparing the sequence space occupied by their LCRs. One fruitful comparison was between the human extracellular matrix and plant cell wall. Each of these have taken a role in the extracellular space, yet they have different chemical compositions, structures, and proteins. The LCR spaces occupied by these proteins are unique for each extracellular assembly, corresponding to differences in the specific interactions and processes required for their formation. While human ECM and plant cell wall proteins both occupy spaces which have a substantial presence of prolines, the specific differences in the regions they occupy give insight into their unique properties. LCRs of ECM proteins occupy the G/P-rich cluster and the presence of glycines in ECM collagen proteins is crucial for tight packing of helices to form the collagen triple helix (*Beck et al., 2000*), which is the basis for higher order assembly of most of the tissues in the human body. On the other hand, while plant cell wall proteins also use polyproline II helices, these P-rich LCRs occupy a different region in the map from LCRs in the ECM. Moreover, plant cell wall proteins contain multiple P-rich LCR compositions which delineate between extensins and other proline-rich cell wall proteins. Such differences, in whether or not the contiguous prolines are interrupted, have been proposed to explain the origins of plant cell wall proteins with different properties (*Kieliszewski and Lamport, 1994*; *Lamport et al., 2011*), supporting the idea that functional divergence of LCRs can occur through relatively local differences in sequence space. This view of LCR sequence space captures key sequence determinants of LCR function in higher order assemblies and highlights that even small differences in LCR sequence space may have meaningful biological consequences.

As the functions of other regions of LCR sequence space are uncovered or mapped, such as the teleost-specific T/H-rich cluster we identified, species with different higher order assemblies and cellular organizations may be found to occupy similar or different spaces. By viewing these LCRs from the perspective of higher order assembly, we suggest that the principles of assembly may be the principles which explain the disparate functions of LCRs. For now, we can only speculate that disparate LCR functions may not be isolated processes, but different regions across a unified LCR space.

## Materials and methods

**Key resources table**

| Reagent type (species) or resource | Designation | Source or reference | Identifiers | Additional information |
|---|---|---|---|---|
| Cell line (Human, Female) | HeLa | ATCC | | Tested negative for Mycoplasma |
| Antibody | Anti-MPHOSPH10 (MPP10) (rabbit polyclonal) | Novus Biologicals | NBP1-84341 | (1:100) |
| Antibody | Anti-Fibrillarin (mouse monoclonal) | EMD Millipore | MABE1154 | (1:100) |
| Antibody | Anti-POLR1A (rabbit polyclonal) | Novus Biologicals | NBP2-56122 | (1:100) |
| Antibody | Anti-UBTF (rabbit polyclonal) | Novus Biologicals | NBP1-82545 | (1:100) |
| Antibody | Anti-rabbit IgG Alexa 647 (goat polyclonal) | Invitrogen | A-27040 | (1:1000) |
| Antibody | Anti-mouse IgG Alexa 594 (goat polyclonal) | Invitrogen | A-11032 | (1:1000) |

*Continued on next page*

*Continued*

| Reagent type (species) or resource | Designation | Source or reference | Identifiers | Additional information |
|---|---|---|---|---|
| Recombinant DNA reagent | RPA43 WT; pcDNA3.1(+) meGFP - RPA43 | This paper | RP104 | Human expression plasmid |
| Recombinant DNA reagent | RPA43 ΔK1,2,3; pcDNA3.1(+) meGFP - RPA43 (ΔK223-P234, P274-Q284, H306-H315) | This paper | RP105 | Human expression plasmid |
| Recombinant DNA reagent | RPA43 ΔK3; pcDNA3.1(+) meGFP - RPA43 (ΔH306-H315) | This paper | RP108 | Human expression plasmid |
| Recombinant DNA reagent | RPA43 ΔK1,2; pcDNA3.1(+) meGFP - RPA43 (ΔK223-P234, P274-Q284) | This paper | RP109 | Human expression plasmid |
| Recombinant DNA reagent | RPA43 ΔK1,3; pcDNA3.1(+) meGFP - RPA43 (ΔK223-P234, H306-H315) | This paper | RP110 | Human expression plasmid |
| Recombinant DNA reagent | RPA43 ΔK2,3; pcDNA3.1(+) meGFP - RPA43 (ΔP274-Q284, H306-H315) | This paper | RP111 | Human expression plasmid |
| Recombinant DNA reagent | RPA43 ΔK1; pcDNA3.1(+) meGFP - RPA43 (ΔK223-P234) | This paper | RP112 | Human expression plasmid |
| Recombinant DNA reagent | RPA43 ΔK2; pcDNA3.1(+) meGFP - RPA43 (ΔP274-Q284) | This paper | RP113 | Human expression plasmid |
| Recombinant DNA reagent | TCOF WT; pcDNA3.1(+) meGFP - TCOF | This paper | RP133 | Human expression plasmid |
| Recombinant DNA reagent | TCOF ΔK; pcDNA3.1(+) meGFP - TCOF (ΔK1390-K1406, K1438-K1468, K1476-K1483) | This paper | RP157 | Human expression plasmid |
| Recombinant DNA reagent | Recombinant RPA43 C-terminus; pGEX6p1 GST-SBP-eGFP - RPA43 (E209-end) | This paper | RP106 | Bacterial expression plasmid |
| Recombinant DNA reagent | Recombinant RPA43 C-terminus ΔK1,2,3; pGEX6p1 GST-SBP-eGFP - RPA43 (E209-end) (ΔK223-P234, P274-Q284, H306-H315) | This paper | RP107 | Bacterial expression plasmid |
| Software, algorithm | NumPy | NumPy | RRID:SCR_008633 | 1.20.1 |
| Software, algorithm | BioPython | BioPython | RRID:SCR_007173 | 1.78 |
| Software, algorithm | Pandas | Pandas | RRID:SCR_018214 | 1.2.3 |
| Software, algorithm | Mahotas | Mahotas; https://mahotas.readthedocs.io/en/latest/ | n/a | 1.4.11 |
| Software, algorithm | SciPy | SciPy | RRID:SCR_008058 | 1.6.2 |
| Software, algorithm | Scanpy | Scanpy | RRID:SCR_018139 | 1.6.2 |
| Software, algorithm | AnnData | AnnData | RRID:SCR_018209 | 0.7.5 |
| Software, algorithm | NetworkX | NetworkX | RRID:SCR_016864 | 2.3 |
| Software, algorithm | Matplotlib | Matplotlib | RRID:SCR_008624 | 3.4.1 |
| Software, algorithm | Seaborn | Seaborn | RRID:SCR_018132 | 0.11.1 |
| Software, algorithm | Dotplot pipeline | This paper; https://doi.org/10.5281/zenodo.6568194 | https://doi.org/10.5281/zenodo.6568194 | |
| Software, algorithm | SEG | NCBI; ftp://ftp.ncbi.nlm.nih.gov/pub/seg/seg/ | n/a | |
| Software, algorithm | fLPS | PMID:29132292 | PMID:29132292 | |

## Experimental Methods

### Plasmids

Note: All constructs (both Mammalian expression and bacterial expression) contain a GSAAGGSG peptide linker between GFP and the protein of interest.

## Mammalian expression constructs

| Plasmid | Source | Identifier |
|---|---|---|
| pcDNA3.1(+) meGFP - RPA43 | This paper | RP104 (RPA43 WT) |
| pcDNA3.1(+) meGFP - RPA43 (ΔK223-P234, P274-Q284, H306-H315) | This paper | RP105 (RPA43 ΔK1,2,3) |
| pcDNA3.1(+) meGFP - RPA43 (ΔH306-H315) | This paper | RP108 (RPA43 ΔK3) |
| pcDNA3.1(+) meGFP - RPA43 (ΔK223-P234, P274-Q284) | This paper | RP109 (RPA43 ΔK1,2) |
| pcDNA3.1(+) meGFP - RPA43 (ΔK223-P234, H306-H315) | This paper | RP110 (RPA43 ΔK1,3) |
| pcDNA3.1(+) meGFP - RPA43 (ΔP274-Q284, H306-H315) | This paper | RP111 (RPA43 ΔK2,3) |
| pcDNA3.1(+) meGFP - RPA43 (ΔK223-P234) | This paper | RP112 (RPA43 ΔK1) |
| pcDNA3.1(+) meGFP - RPA43 (ΔP274-Q284) | This paper | RP113 (RPA43 ΔK2) |
| pcDNA3.1(+) meGFP - TCOF | This paper | RP133 (TCOF WT) |
| pcDNA3.1(+) meGFP - TCOF (ΔK1390-K1406, K1438-K1468, K1476-K1483) | This paper | RP157 (TCOF ΔK) |

## Bacterial expression and purification constructs

| Plasmid | Source | Identifier |
|---|---|---|
| pGEX6p1 GST-SBP-eGFP - RPA43 (E209-end) | This paper | RP106 (RPA43 C-term WT) |
| pGEX6p1 GST-SBP-eGFP - RPA43 (E209-end) (ΔK223-P234, P274-Q284, H306-H315) | This paper | RP107 (RPA43 C-term ΔK1,2,3) |

## Cell lines
HeLa cells were obtained from ATCC. Cells tested negative for mycoplasma.

## Cell Culture
HeLa cells were cultured in 5% CO2 on cell culture-treated 10 cm plates (Genesee Scientific, 25–202) in Dulbecco's Modified Eagle Medium (DMEM, Genesee Scientific, 25–500) supplemented with 10% Fetal bovine serum (FBS, Gemini Bio-products, 100–106) and 1% Penicillin/Streptomycin (Gibco, 10378–016). Cells were split 1:10 every 3 days by using trypsin (Gibco, 25200072).

## Protein purification
All protein purification constructs used were cloned into a version of the pGEX-6P-1 plasmid modified to include eGFP followed by a GSAAGGSG peptide linker. All RPA43 C-terminal (amino acid positions 209–338) fragments were fused to the C-terminus of this linker. After sequence verification, plasmids encoding the final constructs were transformed into 20 μL of Rosetta (DE3) competent cells (EMD Millipore, 70954) and grown overnight at 37 °C in 5 mL LB containing 100 μg/mL Ampicillin (Fisher Scientific, BP1760) and 34 μg/mL Chloramphenicol (Fisher Scientific, BP904-100). Overnight cultures were added to 250 mL of Superbroth containing Ampicillin and Chloramphenicol (same concentrations as above) and grown at 37 °C to an $OD_{600}$ ~0.6–0.8. Cultures were cooled to 4 °C, expression of proteins was induced by the addition of IPTG to a final concentration of 0.5 mM, and cultures were grown on a shaker overnight at 15 °C. Cells were pelleted by centrifugation for 35 min at 9790 x g at 4 °C, and pellets were frozen at –80 °C.

Pellets were thawed and lysed on ice in 15 mL lysis buffer containing freshly added lysozyme and benzonase prepared according to manufacturer instructions (Qiagen Qproteome Bacterial Protein Prep Kit, Cat. No. 37900), 1 mM PMSF (ThermoFisher Scientific, 36978), and 1.5 cOmplete mini EDTA-free protease inhibitor cocktail tablets (Millipore Sigma, 11836170001) per 250 mL culture. Lysates were incubated on ice for 20 min with occasional inversion, and sonicated for 5 cycles (30 s on, 30 s off, high intensity) on a Bioruptor 300 at 4–6 °C. Cellular debris and unlysed cells were pelleted by centrifugation for 30 min at 12,000 x g at 4 °C.

Cleared lysates were syringe filtered (Pall Life Sciences, Product ID 4187) and added to 0.625 mL of glutathione-sepharose beads (GE Healthcare, GE17-0756), which were pre-equilibrated in equilibration buffer (1 X PBS, 250 mM NaCl, 0.1% Tween-20) by performing four 10 mL washes for 5 min each with end-over-end rotation at 4 °C. After addition of filtered lysates, beads were incubated for 2 hr at 4 °C on an end-over-end rotator. Beads were centrifuged at 500 x g for 2 min and unbound lysate was removed. Beads were washed three times for 10 min with 10 mL cold wash buffer (150 mM NaCl, 10 mM MgCl$_2$, 10 mM Na$_2$HPO$_4$, 2 mM ATP) at 4 °C with end-over-end rotation. Three to five 0.5 mL elutions were performed at 4 °C on a nutator, with freshly prepared elution buffer (100 mM TRIS pH 8, 20 mM reduced glutathione, 5 mM EDTA pH 8, 2 mM ATP), each for 10 minutes. Elutions were collected, concentrated, and subsequently buffer exchanged into protein storage buffer (25 mM Tris pH 7.5, 150 mM KCl, 0.5 mM EDTA, 0.5 mM DTT freshly added, 10% glycerol) using Amicon Ultra-0.5 centrifugal filter units with a 10 kDa cutoff (Millipore Sigma, UFC5010). Protein concentrations were determined, after which proteins were diluted to 100 µM in protein storage buffer, aliquoted, and stored at –80 °C.

## Droplet formation assays

Droplet formation assays were performed in droplet formation buffer (50 mM Tris pH 7.0, 150 mM NaCl), in the presence of a final concentration of 10% PEG-8000 (New England Biolabs, B1004), in a total volume of 12 µL. Droplet formation was initiated by the addition of 1 µL of purified protein (in protein storage buffer) to 11 µL of pre-mixed Droplet formation buffer and PEG-8000 on ice (8.6 µL of Droplet formation buffer +2.4 µL 50% PEG-8000). The final protein concentration in the reaction was 8.3 µM. After the addition of purified protein, the reaction was mixed by pipetting, 10 µL was loaded onto a microscope slide (Fisher Scientific, 12-544-2), and droplets were immediately imaged using a fluorescent microscope (Evos FL) at 40 X magnification. Representative images were chosen for *Figure 2*.

Droplet formation assays were repeated over the course of about 6 months, with each replicate corresponding to the same experiment carried out on different days, using the same preparation of purified protein.

## Immunofluorescence

Glass coverslips (Fisherbrand, 12-545-80) were placed in 24-well plates (Genesee Scientific, 25–107) and coated in 3 µg/mL of fibronectin (EMD Millipore, FC010) for 30 min at room temperature. HeLa cells were seeded in each well at 50,000 cells per well. 24 hr after seeding, the cells were transfected with GFP-tagged protein plasmids using Lipofectamine 2000 (Invitrogen, 11668027). Each well was transfected using 100 ng of plasmid and 1 µL of Lipofectamine 2000 in a total of 50 µL of OptiMEM (Gibco, 31985070) according to the Lipofectamine 2000 instructions. Cells on glass coverslips were collected for immunofluorescence 48 hr after transfection. Cells were collected by washing with 1 x PBS (Genesee Scientific, 25–508) and fixation in 4% paraformaldehyde (PFA) for 15 min at room temperature, followed by another three washes with 1 x PBS. Cells were permeabilized and blocked by incubation in blocking buffer (1% BSA (w/v), 0.1% Triton X-100 (v/v), 1 x PBS) for 1 hr at room temperature. Coverslips were then incubated overnight at 4 °C in a 1:100 dilution of primary antibody (anti-MPP10, Novus Biologicals, NBP1-84341; anti-Fibrillarin, EMD Millipore, MABE1154; anti-POLR1A, Novus Biologicals, NBP2-56122; anti-UBTF, Novus Biologicals, NBP1-82545) in blocking buffer. After 3 washes with blocking buffer, coverslips were incubated for 2 hr in a 1:1000 dilution of secondary antibody (anti-rabbit IgG Alexa 647, Invitrogen, A-27040; anti-mouse IgG Alexa 594, Invitrogen, A-11032). Coverslips were washed three times with blocking buffer, then once with 1 x PBS. For RPA43 experiments, coverslips were then mounted on glass slides using ProLong Diamond antifade mountant with DAPI (Invitrogen, P36962). For TCOF experiments, coverslips were stained with Hoechst 33342 (Thermo Scientific, 62249) for 15 min, then washed twice with 1 x PBS, and mounted on glass slides using ProLong Diamond antifade mountant (Invitrogen, P36961). Slides were sealed using clear nail polish, allowed to dry, and stored at 4 °C.

For all immunofluorescence images other than *Figure 4—figure supplement 1D*, slides were imaged on a DeltaVision TIRF microscope using 100 X oil immersion objective lens. Raw images were deconvolved, from which a max projection image was generated. Deconvolution and max projection were performed using Deltavision SoftWoRx software. For *Figure 4—figure supplement 1D*, slides

were imaged on an Olympus FV1200 Laser Scanning confocal microscope. In all cases, displayed images were scaled such that the spatial distribution of signal was representative.

The same set of exposure conditions (one exposure per channel) was used across all slides within the same experiment. Image analysis was performed using Fiji (https://imagej.net/software/fiji/). For each transfected construct, representative cells were chosen. Cells that were excluded were cells that were not appreciably transfected, and cells that highly overexpressed the transfected constructs.

The immunofluorescence experiments were performed multiple times over the course of about 1 year, with each replicate corresponding to the same experiment carried out on different days.

## External data
### Proteome datasets
Proteomes were downloaded from UniProt for all species analyzed (see table below). Every proteome was greater than 90% complete based on Benchmarking Universal Single-Copy Ortholog (BUSCO) assessment score for proteome completeness. One protein sequence was downloaded per gene in FASTA format. Thus, all protein names used in the manuscript are UniProt protein names (i.e. "NUCL" in "NUCL_HUMAN").

| Species | Proteome ID | Date accessed |
| --- | --- | --- |
| *Homo sapiens* | UP000005640 | March 15, 2021 |
| *Mus musculus* | UP000000589 | March 15, 2021 |
| *Danio rerio* | UP000000437 | March 15, 2021 |
| *Drosophila melanogaster* | UP000000803 | March 15, 2021 |
| *Caenorhabditis elegans* | UP000001940 | March 15, 2021 |
| *Saccharomyces cerevisiae* | UP000002311 | March 15, 2021 |
| *Arabidopsis thaliana* | UP000006548 | March 15, 2021 |
| *Escherichia coli* | UP000000625 | March 15, 2021 |
| *Electrophorus electricus* | UP000314983 | July 14, 2021 |
| *Esox lucius* | UP000265140 | July 14, 2021 |
| *Lepisosteus oculatus* | UP000018468 | August 5, 2021 |
| *Salmo salar* | UP000087266 | July 14, 2021 |
| *Takifugu rubripes* | UP000005226 | July 13, 2021 |

### Higher order assembly annotations
Annotations for higher order assemblies were downloaded from Uniprot, based on their subcellular location annotations. Only entries which were Swiss-Prot reviewed (i.e. entry belongs to the Swiss-Prot section of UniProtKB) were included in the annotations. Annotations were accessed in FASTA format. Annotations for stress granule were taken from a published experiment (*Jain et al., 2016*). Stress granule protein sequences from the "Tier1" list of stress granule proteins were downloaded from UniProt in FASTA format.

| Species | Annotation | Date accessed |
| --- | --- | --- |
| *Homo sapiens* | Nucleus speckle (SL0186) | September 30, 2020 |
| | Extracellular matrix (SL0111) | October 27, 2020 |
| | Nucleolus (SL0188) | October 7, 2020 |
| | Nuclear pore complex (SL0185) | May 6, 2021 |
| | Centrosome (SL0048) | April 12, 2021 |
| | PML body (SL0465) | October 8, 2020 |

*Continued on next page*

*Continued*

| Species | Annotation | Date accessed |
|---|---|---|
| | Stress granule (*Jain et al., 2016*) | May 18, 2021 |
| *Arabidopsis thaliana* | Nucleus speckle (SL0186) | August 5, 2021 |
| | Cell wall (SL0041) | June 17, 2021 |
| *Saccharomyces cerevisiae* | Nucleolus (SL0188) | March 16, 2021 |
| | Cell wall (SL0041) | June 17, 2021 |

## Core approach

See *Figure 1D* for overview and flowchart. All code was written in Python 3 and run on Google Colaboratory or the Luria server at MIT. Python modules used were NumPy (1.20.1), BioPython (1.78), Pandas (1.2.3), Mahotas (1.4.11), SciPy (1.6.2), Scanpy (1.7.2), AnnData (0.7.5), NetworkX (2.3), Matplotlib (3.4.1), Seaborn (0.11.1). Code, dotplot module outputs, and other relevant files can be found on zenodo (https://doi.org/10.5281/zenodo.6568194).

### Dotplot generation (Module 1)

Self-comparison dotplots of every protein sequence of every proteome were generated using a custom implementation to make dotplots in which every identically matching amino acid equals 1 and every non-matching position equals 0. For each dotplot, protein sequences from the proteome FASTA file were integer-encoded such that each of the 20 amino acids corresponds to a unique integer from 1 to 20, inclusive. For the null proteome, length-matched sequences were randomly generated from uniformly distributed integers from 1 to 20. A total of two arrays of this sequence x N, row-wise and column-wise, were generated, such that each array was a matrix of size N x N, where N is the protein sequence length. The two matrices were subtracted such that any identical amino acid matches equaled 0 and non-matches were non-zero. The final dotplot matrix was generated by replacing any 0 values with 1 and replacing any non-zero values with 0. Dotplot matrices were saved as temporary files in .npz format using the file saving and compression implementation from NumPy. For images of dotplots, matrices were plotted directly.

### LCR calling (Module 2, part 1)

LCRs were called by identifying high density regions in protein dotplots through classic image processing methods, such as kernel convolution, thresholding, and segmentation (*Figure 1D*).

To identify high density regions in dotplots, we performed kernel convolution on the dotplots with a uniform 10x10 kernel, which calculates a convolved pixel intensity value from 0 to 100 based on the number of dots in that window. This kernel relates to the minimum length of an LCR.

We used the convolved dotplots to determine this 'high density' cutoff to define LCRs. Specifically, we used a false discovery rate (FDR)-based approach to threshold the convolved pixel intensities in a way that reliably identifies high density regions and treats the same sequence similarly regardless of the proteome it comes from. For a given proteome, we generated a background model by simulating an equally sized, length-matched 'null proteome', whose sequences were generated from a uniform amino acid distribution. Using a uniform amino acid distribution for the null proteome minimizes proteome-specific effects on whether a sequence is considered to contribute to a region of high density in a dotplot (See Appendix 1 for a full explanation). Moreover, matching the lengths of the proteomes accounts for differences in the length distributions of proteins in different proteomes. We compared the distribution of convolved pixel intensities from all convolved dotplots of proteins in the real proteome, with those from the null proteome, and identified the lowest convolved pixel intensity which satisfied a stringent FDR of 0.002 (*Figure 1D*, *Figure 5—figure supplement 1*). FDR was defined by the number of pixels from the null set which pass the threshold divided by the total number of pixels which pass the threshold (from the real and null sets combined). This threshold was then applied to every protein in the proteome to generate segmented dotplots, in which high-density regions (referred to as segmented regions) had values of 1 while other regions had values of 0. The

positions from –4 and +5 of the boundaries of the segmented regions were included as the start and stop of the LCR to account for the convolution kernel size. The exception to this was LCRs which existed within that distance from the start or stop of a protein, in which the protein start or stop was designated the start or stop accordingly. Only segmented regions which intersected with the diagonal were called as LCRs.

## LCR type and copy number determination (Module 2, part 2)

To computationally determine the types of LCRs and the copy number for each type, we determined the presence of segmented regions at the intersection between called LCRs in the segmented dotplot (*Figure 1D and E*). For each protein, we represented the LCRs as a network in which the LCRs were nodes and intersections between LCRs were edges (*Figure 1D and E*, *Figure 1—figure supplement 4*). The total number of nodes equals the total number of LCRs in the protein. The number of connected components of this network equals the number of distinct LCR types in the protein. Therefore, the number of nodes within a given connected component equals the number of LCRs of that type. NetworkX (version 2.3) was used to calculate these values, and plot the network representation of LCR relationships within proteins.

## Entropy calculation, random length-matched sequence sampling

Shannon entropy was calculated for each LCR sequence and a length-matched sequence which was randomly sampled from the respective proteome. Random length-matched sequence sampling was done by indexing the position of all proteins in the proteome from 1 to the length of the proteome (i.e. the sum of lengths of all proteins), and randomly selecting a position between 1 and the length of the proteome minus the length of the sequence of interest. The randomly sampled sequence was the sequence of the matched length, starting at the selected position. Shannon entropy for both the LCR and randomly sampled sequence was calculated using Scipy's implementation.

## Other LCR calling methods (SEG/fLPS)

LCRs were called with other methods, SEG (*Wootton and Federhen, 1993*) and fLPS (*Harrison, 2017*) for comparison.

SEG was run on the human proteome using 'default', 'intermediate', and 'strict' settings, as defined by the PLAtform of TOols for LOw COmplexity (PlaToLoCo) (*Jarnot et al., 2020*). Settings used from PlaToLoCo (http://platoloco.aei.polsl.pl/#!/help, accessed May 20, 2021) are restated here for completeness. 'Default': $W$=12, $K1$=2.2, $K2$=2.5; 'Intermediate': $W$=15, $K1$=1.9, $K2$=2.5 (*Huntley and Golding, 2002*); 'Strict': $W$=15, $K1$=1.5, $K2$=1.8 (*Radó-Trilla and Albà, 2012*). From the output, we extracted the LCR coordinates for use in downstream entropy calculations. SEG was downloaded from ftp://ftp.ncbi.nlm.nih.gov/pub/seg/seg/ on May 20, 2021.

fLPS was run on the human proteome using 'default' and 'strict' settings, as defined by the PLAtform of TOols for LOw COmplexity (PlaToLoCo) (*Jarnot et al., 2020*), with a uniform background amino acid composition. Settings used from PlaToLoCo (http://platoloco.aei.polsl.pl/#!/help, accessed May 20, 2021) are restated here for completeness. 'default: m=15, M=500, t=0.001, c=equal; 'strict': m=5, M=25, t=0.00001, c=equal. From the output of fLPS, 'whole' rows were dropped in order to remove LCR calls covering the full length of a protein, which obscured LCR calls of subsequences of proteins. We then extracted the LCR coordinates for use in downstream entropy calculations. fLPS was downloaded from:https://github.com/pmharrison/flps on May 20, 2021 (*Harrison, 2021*).

## Generation of LCR maps (UMAP dimensionality reduction, Leiden clustering)

LCR maps contained a two-dimensional representation of different LCR amino acid compositions. For each LCR in the proteome, the amino acid composition was calculated as the frequency of each amino acid in the LCR, and was represented as a vector in 20-dimensional space. The 20-dimensional vectors of all LCRs were saved in AnnData format as an array in which rows were LCRs and columns were the amino acid frequencies. LCR maps were generated by dimensionality reduction from 20 to 2 dimensions using Scanpy's implementation of UMAP (random_state = 73, n_components = 2; n_neighbors = 200 for *Figure 5*. n_neighbors = default for *Figures 3A and 6D*; *McInnes and Healy, 2020*; *Wolf et al., 2018*). Amino acid distributions on the LCR map were generated by coloring each point on a color scale corresponding to the frequency of the amino acid represented. Leiden clustering was

performed using Scanpy (random_state = 73). For each Leiden cluster, the most represented amino acids in each cluster was manually, but systematically, determined by comparing to the UMAPs with the single amino acid distributions (*Figure 3—figure supplement 1*, *Figure 5—figure supplement 2*) and annotating based on the highest frequency amino acid(s) in a given cluster.

## Annotation of LCR maps (higher order assemblies, biophysical predictions)

Annotation of LCRs belonging to higher order assemblies (see table above) was done by adding annotations to the AnnData object and coloring the LCRs using Scanpy's plotting implementation. Wilcoxon rank sum (MannWhitneyU) tests for amino acid enrichment in LCRs of higher order assemblies were performed using Scanpy. For the Wilcoxon rank sum tests comparing one annotation against all other LCRs, default settings were used. For the Wilcoxon rank sum tests comparing between two annotation sets of LCRs, one annotation set was set as the reference. Within each comparison, all tests were corrected for multiple testing of amino acids using the Benjamini-Hochberg method.

Biophysical predictions were calculated and mapped for all LCRs. For IUPred2A and ANCHOR2 predictions, which are context dependent, the scores at each position were calculated for full-length proteins in the proteome using a modified version of the official python script (*Mészáros et al., 2018*; https://iupred2a.elte.hu/download_new, accessed May 31, 2021) to allow for batch predictions. LCR positions identified by our dotplot approach were used to extract the corresponding ANCHOR and IUPred2A scores for each position in each LCR. The mean ANCHOR and IUPred2A scores for each LCR were calculated and used to color the UMAP plot. The IUPred2A and ANCHOR2 scoring was run with the default 'long' setting and '-a' to include ANCHOR predictions. Kappa scores (*Das and Pappu, 2013*; *Holehouse et al., 2017*) for mixed-charge distribution were calculated for each LCR using the localCIDER package (version 0.1.19). It should be noted that this approach can be used to color the LCRs on the UMAP with any other LCR-specific metrics.

## Acknowledgements

We thank all members of the Calo lab, as well as Eeshit D Vaishnav, Connor Kenny, Christopher B Burge, Amy E Keating, and David P Bartel for helpful discussions and feedback on the manuscript. We would also like to thank the Swanson Biotechnology Center Microscopy and Barbara K Ostrom (1978) Bioinformatics core facilities in the Koch Institute at MIT.

## Additional information

### Funding

| Funder | Grant reference number | Author |
| --- | --- | --- |
| National Institute of General Medical Sciences | R35GM142634 | Eliezer Calo |
| Ludwig Center for Molecular Oncology | | Nima Jaberi-Lashkari |
| National Institutes of Health | T32GM007287 | Byron Lee |
| Pew Charitable Trusts | | Eliezer Calo |
| National Cancer Institute | P30-CA14051 | Eliezer Calo |

The funders had no role in study design, data collection and interpretation, or the decision to submit the work for publication.

### Author contributions

Byron Lee, Nima Jaberi-Lashkari, Conceptualization, Data curation, Software, Formal analysis, Supervision, Validation, Investigation, Visualization, Methodology, Writing - original draft, Project administration, Writing – review and editing; Eliezer Calo, Resources, Supervision, Funding acquisition, Writing – review and editing

### Author ORCIDs
Byron Lee (iD) http://orcid.org/0000-0001-7132-2662
Nima Jaberi-Lashkari (iD) http://orcid.org/0000-0002-0238-9078
Eliezer Calo (iD) http://orcid.org/0000-0002-3006-2742

### Decision letter and Author response
Decision letter https://doi.org/10.7554/eLife.77058.sa1
Author response https://doi.org/10.7554/eLife.77058.sa2

---

## Additional files

### Supplementary files
• Supplementary file 1. p-values for Wilcoxon Rank-Sum Tests. Exact Benjamini-Hochberg corrected p-values for all Wilcoxon Rank-Sum Tests performed in the manuscript are provided, with the corresponding figures indicated. The columns labelled (1-20) correspond to the amino acids as presented in the order within each respective figure.

• Transparent reporting form

### Data availability
The current manuscript is a largely computational study, and much of the code we developed, and files we generate are used across multiple figures. The code, input, and output files can be found on our zenodo repository at: https://doi.org/10.5281/zenodo.6568194. The zenodo link is open access, and a comprehensive README file is included.

The following previously published dataset was used:

| Author(s) | Year | Dataset title | Dataset URL | Database and Identifier |
|---|---|---|---|---|
| Byron L, Nima JL, Eliezer C | 2021 | A unified view of low complexity regions (LCRs) across species | https://doi.org/10.5281/zenodo.6568194 | Zenodo, 10.5281/zenodo.6568194 |

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

# Appendix 1

## Choice of uniform amino acid frequency as null model for FDR calculations

Our approach identifies LCRs and determines their relationships within proteins by identifying high density regions in dotplots based on their convolved pixel intensity, because these high density regions will correspond to LCRs if they lie on the diagonal or intersections between similar LCRs if they lie off of the diagonal. For a given proteome, we generated a background model by simulating dotplots for an equally sized, length-matched 'null proteome', whose sequences are generated from a given null amino acid distribution model. Given this, it is important to note that the null model we select will determine the background density on a dotplot, which is agnostic to the identity of the amino acids being matched.

For the reasons explained below, we deem a uniform amino acid distribution more appropriate than the amino acid distribution of the given proteome in order to achieve our specific goals.

The primary goal of our dotplot analysis is to identify LCRs (sequences of low entropy) and determine their relationships within proteins, such that we can compare LCRs across species. In order to achieve this, our choice of null model must ensure that (1) we accurately identify sequences of low entropy, and (2) we can compare sequences regardless of the proteome it comes from.

A null model based on the amino acid frequencies of a given proteome will not necessarily allow for accurate identification of LCRs (low entropy sequences), nor will it allow the same sequence to be treated similarly in a different proteome. This is illustrated by considering two hypothetical proteomes with different amino acid frequency distributions:

**Proteome A:** relatively even amino acid distribution
**Proteome B:** almost entirely composed of a single amino acid, e.g. Alanine (A).

## Nulls derived from proteome amino acid frequencies fail to call true LCRs

Suppose we compare a null proteome generated from the amino acid frequencies in **Proteome B**, vs. a null proteome generated from a uniform amino acid distribution, and assess how likely a homopolymeric sequence of any amino acid (e.g. XXXXXXXXX) is to be called an LCR given each of these null proteomes (i.e. exceed the 'background' signal to some degree of confidence).

While the convolved pixel intensity of the homopolymeric X sequence is constant, the background convolved pixel intensity will be much higher in the null proteome derived from **Proteome B** vs. the null proteome derived from a uniform amino acid distribution (since its proteins are almost entirely composed of matching amino acids). Thus, despite this homopolymeric sequence being a true LCR, it is unlikely to be called as one when using a null proteome derived from **Proteome B**. In general, the more skewed the amino acid frequencies of a proteome away from uniform, the less sensitive it will be to detection of any true LCRs (sequences of low entropy).

## LCRs called using nulls derived from different proteomes are not comparable

Let us now consider null proteomes derived from the amino acid frequencies of two different proteomes, **Proteome B**, mentioned above, and **Proteome A**, which has a relatively uniform amino acid distribution. Using the same reasoning presented above, a homopolymeric X sequence is much more likely to be called an LCR in **Proteome A**, and not in **Proteome B**. Thus, by defining the background model as the amino acid distribution of a given proteome, a real low complexity sequence may be called an LCR in one proteome but not in the other. This makes it so that the definition of an LCR could be vastly different depending on the background proteome, and would prevent any comparison of LCR sequences.

A null proteome derived from a uniform background model addresses both of the concerns above by (1) achieving maximal sensitivity for real LCRs (sequences of low entropy), and (2) establishing a shared level of 'background' signal in different proteomes.

