## [Editor Report]

This is a valuable paper that discusses a holistic understanding of LCRs of not only individual proteins but also provides a broad perspective of proteomes across many species. The data presented provides solid evidence on how LCR organisation and assemblies may be shared between subcellular compartmentalisation and extracellular organismal structure.

---

## [Decision Letter]

**Decision letter after peer review:**

[Editors’ note: the authors submitted for reconsideration following the decision after peer review. What follows is the decision letter after the first round of review.]

Thank you for submitting the paper "A unified view of LCRs across species" for consideration by *eLife*. Your article has been reviewed by 3 peer reviewers, and the evaluation has been overseen by a Reviewing Editor and a Senior Editor. The following individual involved in review of your submission has agreed to reveal their identity: Sreenivas Chavali (Reviewer #2).

Comments to the Authors:

We are sorry to say that, after consultation with the reviewers, we have decided that this work will not be considered further for publication by *eLife*.

Specifically, the reviewers believe that current data needs to be reanalyzed and additional experimental data are requested. Usually, papers are rejected from *eLife* if a revision would take substantial experimental work. Detailed recommendations for authors are listed below.

*Reviewer #1 (Recommendations for the authors):*

The paper is extremely dense and difficult to read and appears focused on a relatively narrow question. One barrier is that it is poorly linked to both a clear biological background for LCRs and the linked references in the introduction don't clearly talk about the same phenomena. This needs to be expanded to help readers who are not familiar with LCRs understand why this study is important and would help establish that the work will have a deeper impact. It would also make clearer what this manuscript's novel findings are- at present, it is not clear if results add detail to existing views in the literature, or make new discoveries (for example, the arguments about LCRs and localisation/function). Additionally, terms that are not widely used, such as protein valence, need to be defined to be suitable for a broad audience.

A second issue in the writing is that the relationship between the Results sections is unclear. The broad flow of the paper as I read it is that the authors present the method, apply the method to proteomes, experimentally explore one protein, go back to considering proteomes, then compare different proteomes. Specifically, the experimental work is not well justified- it is not clear how that protein was chosen, and how the findings relate to the work presented specifically. It doesn't validate the other findings in a meaningful way and raises questions of valence that don't arise from the analysis. It should be better linked to the rest of the presented work or removed.

To test potential issues with the null the authors should repeat the testing with a null derived from the amino acid composition derived from the individual proteomes under study. This is particularly important in light of some residues apparently rarely appearing in LCRs (e.g. tryptophan).

In the comparisons of the dotplot based method with existing techniques the graphs presented in figure 1 supplement 3 are almost unreadable due to the choice of colours used, particularly in comparing SEG to dotplots (A).

To attempt to remedy issues around the link between LCR contents and function/localisation I recommend the authors:

– Include a justification for the choice of p-value, and confirm in the methods that multiple tests correct took into account both the number of amino acids and the number of function/localisation classes. A discussion of the effect size as well as the p-value would be helpful.

– Use chi-squared and/or permutation tests to test cluster membership (rather than frequency) is more than expected for random perturbation for different amino acids.

– Use a heatmap, possibly with KDEs, to plot the density of specific categories relative to the raw umap. This is particularly important for Figure 3 supp Figure 3 where the points in some categories are so sparse it is hard to see the point distribution. It may be the case that these new plots are quite flat; in this case, it may be better to limit the use of umap plots and focus more heavily on robust statistical tests as umap plots are fundamentally a visualisation technique.

*Reviewer #2 (Recommendations for the authors):*

Although the study presents a unique approach to identify LCRs and exciting insights into LCR sequence space, there are few issues that need to be addressed.

1. It is not clear how the FDR threshold of 0.002, to define the LCRs was chosen. While Figure 1 —figure supplement 2 presents the relevant statistics for different thresholds, it is not intuitive why 0.002 was selected. The authors may consider different representations (distinct colors or dotted vs solid lines) to differentiate the lines corresponding to the thresholds in the supplementary figure. Moreover, the thresholding appears a bit random, as the authors reset the FDR threshold to 0.05 for *E. coli*. Is the threshold species/clade-specific? If so, what factors determine the thresholding. This has to be explicitly discussed for the tool to be used by other researchers in the field.

2. Figure 1: Panels D and E shall be presented first, followed by panels A, B and C. Distinct annotations should be provided for SRRM2 and MUC5A and should be discussed in the figure legends. The authors might consider presenting distinct examples with bars highlighting the similar and distinct types of LCRs in the examples as shown in Figure 1 —figure supplement 4, to allow the readers to appreciate the Dot-plot approach, rightaway. Also, Figure 1 —figure supplement 1 appears to be redundant and can be removed.

3. The discussion on the comparison between SEG, fLPS and the Dot-plot methods is rather limited (lines 209-215). Does the Dot-plot method provide more false-negatives with respect to identifying LCRs?

4. LCRs have been annotated for single amino acids. What was the basis for determining the most prevalent amino acid? The annotation of LCR sequence-space by eye appears arbitrary, hampering reproducibility, should this approach be used by others. Also, the approach does not differentiate between repetitive and non-repetitive LCRs. This must be discussed as limitations in the discussion.

5. On similar lines, physicochemical properties of the amino acids in the LCRs are known to aid formation of higher order assemblies. This implies that though the type of amino acid may vary, but similar physicochemical properties across the LCRs can contribute to similar type of interactions (e.g. G3BP1). Such similarities in the physicochemical properties of the amino acid is not taken into account. Therefore, a protein with multiple LCRs composed of physicochemically similar amino acid types could be treated as "multi distinct" or "multi mixed" in the dot plot method, although the nature of the amino acid type and thereby its mode of function in terms of higher order assemblies, could fall under "multi same" category. The authors should at least discuss these limitations explicitly.

6. The authors may think of some in vitro experiments to test the biological importance of T/H LCRs in Teleosts. Looking into the functions of these proteins might provide insights for designing such experiments. The entire section, as provided, is a descriptive account of the observations, with limited biological insights.

*Reviewer #3 (Recommendations for the authors):*

1. I would like the authors to more fully compare and contrast their approach with a SEG-based approach that extracts LCRs and then compares their composition. I believe they already used SEG to find LCRs, so this should be pretty simple (i.e. generate UMAPS based on the composition of the LCRs identified by SEG vs. their dotplot matrix method).

2. I would like to see the role of codon bias and repeat expansions discussed (and ideally considered, perhaps as an explanation for the bridge sequences)

3. The manuscript felt rather long – I understand the desire to place as many interesting observations in as possible, but as mentioned these are unavoidably cherry-picked, inasmuch as they are interesting and compelling to the authors (and, to this reviewer, I should say – the cherry-picking is unavoidable and not a criticism!). Perhaps instead some of these could be summarized with a table that lists protein, function, LCR, and possible observation? This I think would help avoid readers needing to wade through lots of text to obtain what really amounts to very nice hypotheses, as opposed to specific observations that require open prose. The teleost-specific analysis felt sort of tacked on at the end, and I'm not sure added much (but am happy to be disagreed with). I would like to see the manuscript length reduced, ideally with some of the observations condensed into a table.

4. I would like to more concretely know what (specific) question(s) this work answers. "We don't know much about LCRs" is not a particularly specific motivation, so I'd like to better understand what are the knowledge gap(s) the authors are solving.

5. The authors should provide detailed annotations of all LCRs in an easy-to-access format (both Excel and text-based), and should also I think provide a way to access and make sense of the data shown in the UMAP plots (perhaps with LCRs divided by cluster).

6. Consideration of the papers mentioned on this topic by Eric Ross.

[Editors’ note: further revisions were suggested prior to acceptance, as described below.]

Thank you for resubmitting your work entitled "A unified view of low complexity regions (LCRs) across species" for further consideration by *eLife*. Your revised article has been evaluated by the original reviewwrs, Volker Dötsch (Senior Editor) and a Reviewing Editor.

The manuscript has been improved but there are some remaining issues that need to be addressed, as outlined below:

*Reviewer #3 (Recommendations for the authors):*

This revised version is much improved!

The focus on claiming the dot matrix approach lets one find functionally important LCRs.

Fundamentally, I think the authors are overselling their approach as a way to easily find functionally important subregions in proteins.

I think this method is one way to find amino-acid enriched LCRs. I fundamentally do not believe this approach is somehow magically different from finding low-complexity sequences and then categorizing based on composition after the fact (as one could do with SEG or some other composition-independent metric), or by defining one (or more) residue types to extract contiguous low-complexity subregions enriched for a specific residue (as done by Gutierrez et al).

However, I think this issue can be addressed simply by scaling back the language and making less strong claims regarding what can/cannot currently be done. I recognize these changes were made to address previous critiques, but on my reading of the current version the setup feels like a straw man.

I think the actual way the new intro sets things up is great – I just think the implication that dot matrices are essential for this investigation is unsupported.

To specifically support this suggestion, I have provided some analysis below.

G31BP focus

The initial example focusses on G3BP1, and highlights the fact that there are two flavors of LCRs in G3BP1. The authors then relate those different LCRs back to previously published work to make the point that the dot-plot method can identify functionally distinct subdomains. The authors then state,

"The presence of these compositionally distinct LCRs are critical for the ability of G3BP1 to form stress granules, as the acidic LCRs interact with and inhibit the RGG domain, preventing it from interacting with RNA, a necessary step of stress granule assembly. Thus, by highlighting the relationships between different LCRs, dot plots can provide key insights relevant to protein function."

But the functional insight here didn't come from the dotplots – the functional insight here came from prior work, in which acidic rich and arginine rich subregions were already identified and characterized. The dotplots didn't identify that these regions could interact, nor did it provide any inference regarding the putative functions of these regions. With this in mind I really find the logic here hard to follow; my reading is that the setup is:

1. LCRs are important for biology.

2. Here we provide a new way to find LCRs.

3. By highlighting the relationships between different LCRs, dotplots can provide key insights relevant to protein function.

But this last point really is not what happened – the fact these LCRs are 'different' was not what was used to highlight them as important – instead relating them to prior literature was used to highlight functional roles, and the fact they are 'different' comes from prior work. The chemical logical behind this difference is not explicitly encoded by the dotplot matrix, so (for example) and Arg-rich and Lys-rich LCR would appear equally different.

LCR3 (as the authors define it) is not even particularly arginine rich (4/23 Arg residues) (LRGPGGPRGGLGGGMRGPPRGGM), the RGG domain of G3BP1 contains 11 arginine residues, so most of the arginine's that drive RNA binding actually fall outside of the "arginine-rich LCR3".

In sum, it's unclear to me how the authors' approach relates to the functional investigation of proteins beyond highlighting the low complexity domains that have been previously shown to be important. Does this mean that all 37,342 LCRs in the human proteome should point to functionally-important features? And if not, how is one meant to discriminate between functional vs. non-functional LCRs?

Not to belabor a point, but later the authors then state:

"The examples of dot plots make clear that functional information about LCR type and copy number can be extracted from dot plot matrices "

I still feel like there is a critical step completely missing here; HOW do we extract "functional information" from the dotplots? Are the authors saying that every LCR is functionally important? If yes, in what way? How are we (readers/users) meant to use information for dotplots on previously unstudied proteins to infer functionally important features beyond "If there's an LCR remove it and see if it matters".

The authors go on to say:

"The examples of dot plots make clear that functional information about LCR type and copy number can be extracted from dot plot matrices. However, there currently is not an approach to globally assess these features of LCRs and their functions. While several methods exist for identifying LCRs these methods are unable to determine LCR relationships such as type and copy number"

The way one determines the type and copy number is looking at (1) the amino acid composition of LCRs identified by these other methods and (2) the number of LCRs identified. This is precisely what the authors are doing with their own method.

For example, using SEG on G3BP1 and setting an arbitrary threshold here of 0.45 we find 3 SEG-derived LCRs (highlighted in blue here) which correspond to 144-160 ('PQEESEEEVEEPEERQ'), 192-205 ('EPEPDPEPEPEQE'), and 430-448 ('PGGPRGGLGGGMRGPPRG').

The authors' approach identifies basically the same three regions:

SEG: PQEESEEEVEEPEERQ

DotMatrix: TEPQEESEEEVEEPEERQQ

SEG: EPEPDPEPEPEQE

DotMatrix: AEPEPDPEPEPEQEPVS

SEG: PGGPRGGLGGGMRGPPRG

DotMatrix: LRGPGGPRGGLGGGMRGPPRGGM

Now I want to be clear – I have no particular love for SEG, but my point is simply that to claim there is no way to characterize LCRs that takes type and copy number is demonstrably false.

I would encourage the authors to not oversell the method as some magic bullet, and not pretend one could not do what they have done here with another method. You could. The big difference is people haven't and the actual analysis of proteomes is interesting and I think well written and well-explored.

Codon bias remains undiscussed

When the authors discuss biases and bridging among distinct classes of LCRs, I would, again, STRONGLY encourage them to consider codon biases – can overlapping/related enrichments for specific residues be explained based on codon distance in repetitive and slippy sequences? I recognize the authors state that 'this is out of scope' but… it's not. If the authors are going to find LCRs with specific compositional biases and comment on bridges between distinct classes of LCRs, ignoring a codon-based explanation for these bridges is most definitely within scope.

LCRs are short

One thing I had not appreciated in the original manuscript is that the LCRs are short. In the human proteome, 87% of LCRs are 20 residues or shorter. This to me makes me wonder if LCR is really the right term. The LCRs identified previously have generally been large domains, while here LCRs are functionally of the same length as motifs, should these perhaps be defined as LCMs (low complexity motifs) more

---

## [Author Response]

[Editors’ note: The authors appealed the original decision. What follows is the authors’ response to the first round of review.]

Comments to the Authors:We are sorry to say that, after consultation with the reviewers, we have decided that this work will not be considered further for publication by eLife.Specifically, the reviewers believe that current data needs to be reanalyzed and additional experimental data are requested. Usually, papers are rejected from eLife if a revision would take substantial experimental work. Detailed recommendations for authors are listed below.

We thank the Editors and Reviewers for taking the time to review our manuscript “A unified view of LCRs across species”, and for sharing their comments and concerns. In this document, we have written our full responses to reviewer comments, which includes new analyses and experiments.

The Editor and Reviewers also made clear that additional experimental data was needed to validate hypotheses made in the manuscript. We have provided additional experimental data to address this concern, which can be found in Figure 4 and Figure 4 —figure supplement 1. We have also modified the main text to include descriptions and discussion of these results in the context of the overall manuscript, and updated the Methods to include these experiments.

Briefly, we have added experimental data to demonstrate the role of E-rich LCRs in determining scaffold/client relationships among nucleolar proteins. Figure 4 shows that TCOF, a nucleolar protein with many E-rich LCRs, is a scaffold which can recruit key nucleolar proteins with a low number of E-rich LCRs (see section ‘A unified view of LCRs reveals scaffold-client architecture of E-rich LCR-containing proteins in the nucleolus‘ for more details). These results showcase the utility of our approach to provide meaningful biological insights into the function of LCR-containing proteins.

More broadly, Figure 4 now more completely demonstrates that by providing such a unified view of the sequences, features, and relationships of LCRs, our approach facilitates new discoveries about how LCRs contribute to higher order assemblies, and provides a framework for understanding the role of LCRs in other regions of sequence space.

Reviewer #1 (Recommendations for the authors):The paper is extremely dense and difficult to read and appears focused on a relatively narrow question.

In response to this comment and those of other reviewers, we have attempted to streamline the manuscript by making changes to several sections. We hope that these changes help make the paper easier to read.

In regards to the breadth of the question, we have made changes to the introduction to better convey the breadth of the question we are interested in answering, which we believe will be of interest to many audiences in biology.

One barrier is that it is poorly linked to both a clear biological background for LCRs and the linked references in the introduction don't clearly talk about the same phenomena. This needs to be expanded to help readers who are not familiar with LCRs understand why this study is important and would help establish that the work will have a deeper impact.

As we mentioned in the previous comment, we have made changes to the introduction to better convey the breadth of the question we are interested in answering, which we believe will be of interest to many audiences in biology.

In regards to the linked references in the introduction, we would like to note that part of the background we are introducing is that these phenomena are treated as different processes when there are various similarities between them. In fact, many of the linked references (i.e. Malay *et al.,* 2020; Rauscher and Pomès, 2017) are clearly highlighting that some of the processes, which at face value appear different, indeed have similar underlying processes.

It would also make clearer what this manuscript's novel findings are- at present, it is not clear if results add detail to existing views in the literature, or make new discoveries (for example, the arguments about LCRs and localisation/function).

In response to this comment and one by Reviewer 3, we have reworked the introduction such that it is more explicit about the questions we want to answer and how our findings relate to them. We also made several changes to the beginnings and ends of several sections in the Results to explicitly state how each section and its findings help advance the goal we describe in the introduction, and the field more generally.

Additionally, terms that are not widely used, such as protein valence, need to be defined to be suitable for a broad audience.

We define and provide an introduction to valency in the second paragraph of the Introduction. While this description was by no means exhaustive, we believe that this level of introduction is sufficient to understand the rest of the manuscript, and we point the reader to relevant literature if they wish to read more. If the reviewer believes we have overlooked a critical piece of background on valency which would be helpful to the reader, we would be happy to include it.

A second issue in the writing is that the relationship between the Results sections is unclear. The broad flow of the paper as I read it is that the authors present the method, apply the method to proteomes, experimentally explore one protein, go back to considering proteomes, then compare different proteomes.

We thank the reviewer for this comment. It seems to us based on this comment that the lack of clarity about the flow of the manuscript is due to some lack of clarity about the role that the Figure 2 experiments play in the overall flow of the paper. In our response to the next comment, we hope to provide some clarity on why we believe the manuscript follows a logical flow, and address the next comment of the reviewer.

In addition, we have reworked the introduction such that it is more explicit about the questions we want to answer and how our findings relate to them. We also made several changes to the beginnings and ends of several sections in the Results to explicitly state how each section and its findings help advance the goal we describe in the introduction.

Specifically, the experimental work is not well justified- it is not clear how that protein was chosen, and how the findings relate to the work presented specifically. It doesn't validate the other findings in a meaningful way and raises questions of valence that don't arise from the analysis. It should be better linked to the rest of the presented work or removed.

Questions of LCR copy number (related to valence) and type were a major motivation for how the dotplot method was chosen, and are discussed in the Introduction and at the end of the first section of the Results section. The link between the experimental work in Figure 2 and the flow of the manuscript is that it highlights the utility of our approach in understanding how LCR type and copy number influence protein function, which we have also stated in the text.

In response to this comment, we have made more explicit the rationale for choosing RPA43 at the beginning of this section, and clarify the role that these experiments play in the overall flow of the paper.

To test potential issues with the null the authors should repeat the testing with a null derived from the amino acid composition derived from the individual proteomes under study. This is particularly important in light of some residues apparently rarely appearing in LCRs (e.g. tryptophan).

Thank you for pointing this out. The choice of null proteome was a topic of much discussion between the authors as this work was being performed. While we maintain that the uniform background is the most appropriate, the question from this reviewer and the other reviewers made us realize that a thorough explanation was warranted. For a complete explanation for our choice of this uniform null model, please see the newly added appendix section, Appendix 1.

In the comparisons of the dotplot based method with existing techniques the graphs presented in figure 1 supplement 3 are almost unreadable due to the choice of colours used, particularly in comparing SEG to dotplots (A).

This had escaped our notice, thank you for pointing this out. To improve readability, we have changed the figures to not rely only on the color of the lines. For Figure 1 —figure supplement 3, we have improved visibility by changing the line type, increasing the contrast between colors, and increasing the spacing in the legend in the panels to make the distinction between the groups more clear. Similar adjustments to improve clarity have also been made in other figures with similar plots (Figure 1 —figure supplement 2, and Figure 5 —figure supplement 1).

To attempt to remedy issues around the link between LCR contents and function/localisation I recommend the authors:– Include a justification for the choice of p-value, and confirm in the methods that multiple tests correct took into account both the number of amino acids and the number of function/localisation classes. A discussion of the effect size as well as the p-value would be helpful.

P-value justification:

The p-value threshold for coloring the bars (10^-3^) was chosen arbitrarily as one which we deemed sufficiently stringent. It should be noted that this cutoff is only used to color the bars in the figures containing Wilcoxon-rank sum tests. All of the data (including those with p-values that do not satisfy this threshold) are clearly presented in the same figures, and exact p-values are provided as a supplemental table.

Test correction:

For each annotation set (e.g. nucleolus, nuclear speckle, ECM, etc), a Benjamini-Hochberg multiple test correction took into account the number of amino acids (20). In response to this concern, we have updated the Methods section titled ‘Annotation of LCR maps (higher order assemblies, biophysical predictions)’ to make this explicit.

With respect to correcting based on annotations, we did not perform these tests for an exhaustive list of annotation sets. We have not corrected for the number of annotation sets we tested throughout different sections of the manuscript, which was fourteen in total (see Lee_Jaberi_pvalues.xls). Even a stringent Bonferroni correction would affect these p-values by about one order of magnitude. In the most conservative case where something had a p-value of 10^-3^ before this correction (most had p-values much lower than this), its corrected p-value would be ~10^-2^, which we still deem acceptable.

We should also note that some of these amino acid frequency biases have been documented and experimentally validated by other groups, as we mention and reference in the manuscript.

Effect size:

With respect to the effect size, we would like to provide an example to demonstrate why apparently small effect sizes in average frequency do not affect our conclusions. As we mentioned in one of our earlier responses, many of the proteins in nucleoli have multiple LCRs. While K-rich ones are enriched in nucleolar proteins, not every LCR of every nucleolar protein will be K-rich. Thus, the distribution of K frequency is not unimodal. The many LCRs that have few or no K’s will result in a low mean K frequency in nucleolar LCRs, even though a subset of nucleolar LCRs have very high K frequencies. This is illustrated by looking at the K-rich LCR frequency among all true nucleolar LCRs vs the K frequencies of an equally sized random sample of LCRs. While the means of these two samples may not be incredibly different (giving the illusion of a small effect size), the difference in their distributions is very apparent. Real nucleolar proteins contain many more LCRs with a high K frequency.

**Author response image 1. sa2fig1:** 

– Use chi-squared and/or permutation tests to test cluster membership (rather than frequency) is more than expected for random perturbation for different amino acids.

While we do assign labels to clusters based on the dominant amino acid/s in each cluster, these clusters are only intended to serve as guides when we refer to different regions of the map (as we have stated in the text). While we had initially considered performing analyses on clusters of the UMAP, we decided against it for several reasons which we outline below.

1) We would like our conclusions to not be sensitive to the specific UMAP/clustering parameters used. Thus, we do not perform any analyses on cluster membership as these conclusions may not be robust to different UMAP/clustering settings. Instead, we chose to keep our analyses to intrinsic features of the LCRs themselves (e.g. amino acid frequency), such that our conclusions would be robust.

2) In the case of this UMAP (as opposed to UMAPs based on single-cell sequencing data for example), cluster membership has no additional meaning beyond the amino acid frequencies of the LCRs within it. This is in contrast to UMAPs generated from single-cell sequencing data, where clusters may correspond to certain cell types. Thus, it is unclear what would be gained by performing analyses using clusters.

Thus, while we don’t believe that performing these tests on Leiden clusters would be particularly informative, we instead attempted to address the reviewer’s concern by performing permutation tests to test the probability of obtaining amino acid frequencies greater than the unpermuted set by chance. We did this by randomly assigning the same number of nucleolar ‘labels’ to LCRs, and determining the mean frequency of specific amino acids in the assigned nucleolar group. We did this resampling 10,000 times, and plotted the results below in grey, and indicated the unpermuted amino acid frequency with a vertical red line.

As can be seen in Author response image 2, the permutation test results match well with the results from the Wilcoxon rank-sum tests for enrichment of certain amino acids in LCRs. The example shown is a permutation test for methionine (M) and lysine (K) for LCRs of proteins in the nucleolus. From these permutation tests, it is quite likely that the distribution of methionine frequency of nucleolar LCRs could be arrived at by chance (p = 0.0945), which is consistent with the lack of methionine enrichment observed by Wilcoxon rank-sum test. On the other hand, the lysine frequency distribution seen in LCRs of nucleolar proteins is incredibly unlikely to occur by chance (p < 10^-4^; note this is limited by n permutations), which is also consistent with the lysine enrichment we observe by Wilcoxon rank-sum test.

**Author response image 2. sa2fig2:** Permutation Tests.

We hope that these results illustrate the robustness of the enrichments we observe by Wilcoxon rank-sum tests presented throughout the manuscript.

– Use a heatmap, possibly with KDEs, to plot the density of specific categories relative to the raw umap. This is particularly important for Figure 3 supp Figure 3 where the points in some categories are so sparse it is hard to see the point distribution. It may be the case that these new plots are quite flat; in this case, it may be better to limit the use of umap plots and focus more heavily on robust statistical tests as umap plots are fundamentally a visualisation technique.

We appreciate the reviewer’s comment. We wish to clarify that any claims we make about enrichment of specific amino acids in LCRs of certain higher order assemblies are based on Wilcoxon rank-sum tests, and not a visual interpretation of the UMAPs. These UMAPs are only provided to help see where LCRs lie in the sequence space. The calculations for the statistical tests are independent of the UMAPs. Given that we do not make conclusions based on the UMAP, we believe that the use of statistical tests are more appropriate in demonstrating the enrichments we observe, than a heatmap.

Reviewer #2 (Recommendations for the authors):Although the study presents a unique approach to identify LCRs and exciting insights into LCR sequence space, there are few issues that need to be addressed.1. It is not clear how the FDR threshold of 0.002, to define the LCRs was chosen. While Figure 1 —figure supplement 2 presents the relevant statistics for different thresholds, it is not intuitive why 0.002 was selected. The authors may consider different representations (distinct colors or dotted vs solid lines) to differentiate the lines corresponding to the thresholds in the supplementary figure. Moreover, the thresholding appears a bit random, as the authors reset the FDR threshold to 0.05 for *E. coli*. Is the threshold species/clade-specific? If so, what factors determine the thresholding. This has to be explicitly discussed for the tool to be used by other researchers in the field.

We thank the reviewer for pointing this out. Indeed, our intent is for our approach to be widely accessible to the broader scientific community.

Our choice of FDR threshold was chosen to ensure that the sequences identified were truly LCRs (i.e. low entropy sequences) and that we could maximally identify the LCRs in the proteome. These goals have an inherent tradeoff in that minimizing entropy will be more stringent and therefore reduce the number of LCRs identified. For this reason, out of the thresholds we tested, the FDR of 0.002 was chosen. This can be seen by comparing to the more stringent threshold, 0.001, which identified ~8900 fewer LCRs (Figure 1 —figure supplement 1C), while having little effect on the entropy of the sequences (Figure 1 —figure supplement 1D).

We would also like to note that while this was our reasoning for selecting an FDR of 0.002, the FDR threshold can be set depending on the purpose of the analysis. For example, a researcher who is interested in the lowest entropy sequences, but does not need a large number of them for their analysis may decide to be more stringent with the FDR threshold.

Similar reasoning can be used regarding the *E. coli* threshold, in which we relaxed the FDR threshold. As can be seen in Figure 5 —figure supplement 1C, the sequences identified in *E. coli* at an FDR of 0.05 were still low in entropy, and this less stringent threshold contained LCRs such that further analysis was possible. However, given this difference in FDR, we intentionally do not make strong claims about LCRs in *E. coli*, but included this to be as transparent about the process as possible.

2. Figure 1: Panels D and E shall be presented first, followed by panels A, B and C. Distinct annotations should be provided for SRRM2 and MUC5A and should be discussed in the figure legends. The authors might consider presenting distinct examples with bars highlighting the similar and distinct types of LCRs in the examples as shown in Figure 1 —figure supplement 4, to allow the readers to appreciate the Dot-plot approach, rightaway. Also, Figure 1 —figure supplement 1 appears to be redundant and can be removed.

The reason we chose to present Figure 1A, B, and C before Figure 1D and E, is that Figure 1A, B, and C are images of raw dotplots, and have not been processed by our computational pipeline. We realized after reading this comment that our annotations on the dotplot could be incorrectly interpreted as automated annotations by the pipeline. This is not the case. We sought to simply point out that the information we are interested in for this manuscript (LCR sequences, type, and copy number) are all found in a dotplot representation of a sequence. To help avoid this misunderstanding, we have removed the colored-box annotations on the dotplot, and instead simply point to regions that we refer to in the text. Similarly, we have added arrows to annotate SRRM2 and MUC5A, such that readers can appreciate the information found in dotplots, even without the use of the pipeline. Because these dotplots were not processed with the pipeline, we do not believe the bars showing similar and distinct LCR calls (which are an output of the pipeline) are appropriate here. Hopefully these changes help make clear the message we want to convey in these panels.

We believe that Figure 1 —figure supplement 1 is required to give a reasonable survey of the information which can be found in dotplots, which we attempt to communicate in the section "Dotplots reveal the presence and organization of low complexity regions (LCRs) in proteins". These observations were the motivation for us to make an automated pipeline to generate dotplots and extract this information from dotplots.

3. The discussion on the comparison between SEG, fLPS and the Dot-plot methods is rather limited (lines 209-215). Does the Dot-plot method provide more false-negatives with respect to identifying LCRs?

When considering what to discuss about these methods, we decided to evaluate certain metrics relevant for our downstream analyses: (1) sequence complexity of called LCRs, (2) total number of amino acids in called LCRs, and (3) the ability to determine LCR type and copy number. We have limited our discussion to these criteria because our only claim is that our method is able to determine LCR type and copy number, while being comparable in (1) identifying LCRs of low sequence complexity and (2) a similar number of amino acids in called LCRs. We use these metrics to assess our approach, because in the absence of a 'ground truth' for what is an LCR, it is not clear how to define false negatives.

4. LCRs have been annotated for single amino acids. What was the basis for determining the most prevalent amino acid? The annotation of LCR sequence-space by eye appears arbitrary, hampering reproducibility, should this approach be used by others. Also, the approach does not differentiate between repetitive and non-repetitive LCRs. This must be discussed as limitations in the discussion.

We realize the wording "by eye" was a confusing choice in the text. These annotations were done manually, but systematically, by comparing the human and species UMAP(s) to Figure 3 —figure supplement 1 and Figure 5 —figure supplement 2, respectively. These supplements were generated by plotting the frequency of each amino acid for each point on the UMAP. The regions with the highest frequency for a given amino acid was designated the "most prevalent amino acid". We have updated the text in the Methods section titled ‘Generation of LCR maps (UMAP dimensionality reduction, Leiden clustering)’ to make this explicit.

Regarding repetitiveness, we do not see this as a limitation of the approach because if one can assign a score for ‘repetitiveness’, this can be plotted on the UMAP as we did with biophysical scores in Figure 4 —figure supplement 2 (in the revised version) or Figure 5 —figure supplement 5.

5. On similar lines, physicochemical properties of the amino acids in the LCRs are known to aid formation of higher order assemblies. This implies that though the type of amino acid may vary, but similar physicochemical properties across the LCRs can contribute to similar type of interactions (e.g. G3BP1). Such similarities in the physicochemical properties of the amino acid is not taken into account. Therefore, a protein with multiple LCRs composed of physicochemically similar amino acid types could be treated as "multi distinct" or "multi mixed" in the dot plot method, although the nature of the amino acid type and thereby its mode of function in terms of higher order assemblies, could fall under "multi same" category. The authors should at least discuss these limitations explicitly.

As the reviewer mentions, there are examples where amino acids with similar physical properties are interchangeable. However, given that this equivalence may not generalize to every LCR (e.g. lysines and arginines), we therefore tried to make as few assumptions as possible. If researchers are specifically interested in assigning ‘matches’ in the dotplot to certain sets of similar but non-identical amino acids, our code is publically available and can be modified to achieve this, and the corresponding output of the pipeline would have this equivalence taken into account.

Occasionally, we observe fusion when two hollow spheres make contact, as can be seen in . We also observe spheres which are nested within each other.

Reviewer #3 (Recommendations for the authors):1. I would like the authors to more fully compare and contrast their approach with a SEG-based approach that extracts LCRs and then compares their composition. I believe they already used SEG to find LCRs, so this should be pretty simple (i.e. generate UMAPS based on the composition of the LCRs identified by SEG vs. their dotplot matrix method).

We have done downstream UMAP analysis with LCRs called from SEG. We found that although certain aspects of the dotplot-based LCR UMAP are reflected in the SEG-based LCR UMAP, there is overall worse resolution with default settings, which is likely due to fused LCRs of different compositions. Attempting to improve resolution using more stringent settings comes at the cost of the number of LCRs assessed. We maintain that this comparison is not really the focus of our manuscript. We do not make strong claims about the dotplot matrices being better than SEG, or any other method.

2. I would like to see the role of codon bias and repeat expansions discussed (and ideally considered, perhaps as an explanation for the bridge sequences)

We appreciate the reviewer bringing up these topics on the potential origins of LCRs, such as codon bias and repeat expansions. As we noted in the response above, we believe this discussion is related, but out of the scope of what the paper covers. We were intentional in focusing on the functions of LCRs, and have kept our discussions more directly to that point. For a more detailed explanation, please see our response to the more general comment about repeat expansions above.

3. The manuscript felt rather long – I understand the desire to place as many interesting observations in as possible, but as mentioned these are unavoidably cherry-picked, inasmuch as they are interesting and compelling to the authors (and, to this reviewer, I should say – the cherry-picking is unavoidable and not a criticism!). Perhaps instead some of these could be summarized with a table that lists protein, function, LCR, and possible observation? This I think would help avoid readers needing to wade through lots of text to obtain what really amounts to very nice hypotheses, as opposed to specific observations that require open prose. The teleost-specific analysis felt sort of tacked on at the end, and I'm not sure added much (but am happy to be disagreed with). I would like to see the manuscript length reduced, ideally with some of the observations condensed into a table.

In response to this comment and those of other reviewers, we have attempted to streamline the manuscript by making changes to several sections. We hope that these changes help make the paper easier to read.

With respect to the section on T/H rich LCRs (Figure 6), we acknowledge that the role of this section could have been made more clear in the text. We have updated the beginning of this section to convey its importance.

Briefly, we believe that this figure on the T/H LCRs in Teleosts is a synthesis of the unified view that we presented throughout the paper, by applying what we have learned to understand a previously unknown region of sequence space.

We use information about the sequences, relationships, and features of the LCRs in this space to identify client- and scaffold- like proteins (as we did in Figures 1, 2, and 4) and understand how these sequences may contribute to higher order assembly of these LCRs under some conditions. We gain further insight by making use of our ability to easily compare sequence space occupancy across species (as we did in Figure 5), and demonstrate that occupancy of this sequence space is conserved among teleosts, akin to the conservation of G/P rich LCRs in metazoans. Our observation that this region of sequence space exhibits biochemical and evolutionary signatures of higher order assemblies argues for its biological importance.

4. I would like to more concretely know what (specific) question(s) this work answers. "We don't know much about LCRs" is not a particularly specific motivation, so I'd like to better understand what are the knowledge gap(s) the authors are solving.

In response to this comment and one by Reviewer #1, we have reworked the introduction of the manuscript which we believe is more explicit about the questions we want to answer and how our findings relate to them, and made changes to the end of several sections to explicitly state how each section and its findings help advance the goal we describe in the introduction, and the field more generally.

5. The authors should provide detailed annotations of all LCRs in an easy-to-access format (both Excel and text-based), and should also I think provide a way to access and make sense of the data shown in the UMAP plots (perhaps with LCRs divided by cluster).

We have added excel tables of all LCRs in the UMAPs shown in Figures 3, 5, and 6 to our zenodo repository.

6. Consideration of the papers mentioned on this topic by Eric Ross.

We appreciate the reviewer for noting this related body of work. We have updated the citations to include work from Eric Ross where relevant.

[Editors’ note: what follows is the authors’ response to the second round of review.]

The manuscript has been improved but there are some remaining issues that need to be addressed, as outlined below:Reviewer #3 (Recommendations for the authors):This revised version is much improved!The focus on claiming the dot matrix approach lets one find functionally important LCRs.Fundamentally, I think the authors are overselling their approach as a way to easily find functionally important subregions in proteins.

After considering all of the reviewer’s comments, we believe that many of the reviewer’s comments are a consequence of two major things, both of which may have led to a mismatch between what the reviewer interpreted and what we intended to convey.

1) We’ve come to realize that there are some parts of the text where we can understand why the reviewer would arrive at their conclusions. Given that this is not what we intended to convey, we have made changes such that the text better aligns with our intended message.

2) We also believe that there is a misunderstanding of the line of reasoning we use to argue that a global view of LCR type and copy number provided by a systematic dotplot approach is likely to be fruitful. It is possible that some of our wording may have contributed to this, so we have tried to adjust those parts of the text to avoid confusion.

Our responses below and our revisions aim to address the reviewer’s concerns, clear up potential misunderstandings, and clarify our intended claims and reasoning.

I think this method is one way to find amino-acid enriched LCRs. I fundamentally do not believe this approach is somehow magically different from finding low-complexity sequences and then categorizing based on composition after the fact (as one could do with SEG or some other composition-independent metric), or by defining one (or more) residue types to extract contiguous low-complexity subregions enriched for a specific residue (as done by Gutierrez et al).However, I think this issue can be addressed simply by scaling back the language and making less strong claims regarding what can/cannot currently be done. I recognize these changes were made to address previous critiques, but on my reading of the current version the setup feels like a straw man.I think the actual way the new intro sets things up is great – I just think the implication that dot matrices are essential for this investigation is unsupported.To specifically support this suggestion, I have provided some analysis below.G31BP focusThe initial example focusses on G3BP1, and highlights the fact that there are two flavors of LCRs in G3BP1. The authors then relate those different LCRs back to previously published work to make the point that the dot-plot method can identify functionally distinct subdomains. The authors then state,"The presence of these compositionally distinct LCRs are critical for the ability of G3BP1 to form stress granules, as the acidic LCRs interact with and inhibit the RGG domain, preventing it from interacting with RNA, a necessary step of stress granule assembly. Thus, by highlighting the relationships between different LCRs, dot plots can provide key insights relevant to protein function."But the functional insight here didn't come from the dotplots – the functional insight here came from prior work, in which acidic rich and arginine rich subregions were already identified and characterized. The dotplots didn't identify that these regions could interact, nor did it provide any inference regarding the putative functions of these regions. With this in mind I really find the logic here hard to follow; my reading is that the setup is:1. LCRs are important for biology.2. Here we provide a new way to find LCRs.3. By highlighting the relationships between different LCRs, dotplots can provide key insights relevant to protein function.But this last point really is not what happened – the fact these LCRs are 'different' was not what was used to highlight them as important – instead relating them to prior literature was used to highlight functional roles, and the fact they are 'different' comes from prior work. The chemical logical behind this difference is not explicitly encoded by the dotplot matrix, so (for example) and Arg-rich and Lys-rich LCR would appear equally different.

We appreciate the reviewer's comments, which have been very helpful in understanding how this section may have led to confusion about our logic and claims here. We would like to clarify that the goal of this section ("Dotplots reveal the presence and organization of LCRs in proteins") is to make the argument that developing a systematic approach using dotplots is likely to be fruitful. To make this argument, we use existing literature and single dotplots (not the output of a systematic analysis, which we develop in the next section).

The logic of the argument presented in this section is the following:

1) LCRs are important for biology

2) LCR types and their respective copy numbers can be important for their biological function

→ e.g. LCR types of G3BP1

→ an interesting trend of proteins with many similar LCRs playing scaffolding roles across diverse assemblies

3) Dotplots can identify LCR types and their respective copy numbers.

4) Given points 1, 2, and 3 above, performing a systematic dotplot approach to identify LCR type and copy relationships may provide biological insights. These include, but are not limited to newly identifying proteins which have LCR relationships like known ones such as G3BP1 and may function similarly, identifying potential scaffold proteins on the basis of many similar LCRs (as we do in Figure 4 and 6), or learning more general patterns of LCR type and copy number. This line of reasoning serves as the motivation to develop and implement the systematic approach which we use in the rest of the manuscript.

LCR3 (as the authors define it) is not even particularly arginine rich (4/23 Arg residues) (LRGPGGPRGGLGGGMRGPPRGGM), the RGG domain of G3BP1 contains 11 arginine residues, so most of the arginine's that drive RNA binding actually fall outside of the "arginine-rich LCR3".

We do not refer to LCR3 as ‘arginine-rich’. Our use of "RGG domain" was only to reference the existing literature on this region of G3BP1. We have adjusted the text to make it clear that we are only pointing out that it is different from LCR1 and LCR2 due to the lack of off-diagonal intersections.

In sum, it's unclear to me how the authors' approach relates to the functional investigation of proteins beyond highlighting the low complexity domains that have been previously shown to be important.

If by "approach" the reviewer is specifically referring to the use of dotplots in the G3BP1 section, we agree that in certain places the wording does not align with what we intended and that there may be some confusion there. For clarifications, please see the logic of the argument above. We have made changes in the text to address this concern.

If the reviewer is referring to the rest of the manuscript, we respectfully disagree on the basis of the results we show after this section. We believe that the results shown in the rest of the manuscript show the value of our approach in highlighting the function of low complexity regions that have not been previously shown to be important (see Figures 2 and 4).

Does this mean that all 37,342 LCRs in the human proteome should point to functionally-important features? And if not, how is one meant to discriminate between functional vs. non-functional LCRs?

The goal is not to state that every called LCR will be functional. In the rest of the manuscript (Figures 2, 4, 5), we point to ways to assess whether or not certain LCRs may be functional through copy number analysis (Figure 2 and 4), and conservation (Figure 5).

Not to belabor a point, but later the authors then state:"The examples of dot plots make clear that functional information about LCR type and copy number can be extracted from dot plot matrices "I still feel like there is a critical step completely missing here; HOW do we extract "functional information" from the dotplots?

We can understand how the wording in this sentence would be misleading. We have adjusted the text in this sentence, and removed the word ‘functional’. Our intention is to say that dotplots can extract LCR type and copy number information, and that these LCR relationships may be functional.

Are the authors saying that every LCR is functionally important? If yes, in what way? How are we (readers/users) meant to use information for dotplots on previously unstudied proteins to infer functionally important features beyond "If there's an LCR remove it and see if it matters".

The goal is not to state that every called LCR will be functional, but that the ability of dotplots to capture LCR sequences, types, and copy numbers may provide insight into function, as described in the outline of our logic above. Given this, in the rest of the manuscript (Figures 2, 4, 5), we go on to demonstrate copy number analysis (Figure 2 and 4), and conservation (Figure 5) are ways to assess whether or not certain LCRs may be functional.

The authors go on to say:"The examples of dot plots make clear that functional information about LCR type and copy number can be extracted from dot plot matrices. However, there currently is not an approach to globally assess these features of LCRs and their functions. While several methods exist for identifying LCRs these methods are unable to determine LCR relationships such as type and copy number"The way one determines the type and copy number is looking at (1) the amino acid composition of LCRs identified by these other methods and (2) the number of LCRs identified. This is precisely what the authors are doing with their own method.

We’d like to make two things clear here:

A) Determining LCR type using dotplots does not involve an assignment of composition type, followed by counting. Rather, it compares the LCRs directly in order to determine "similar" or "distinct" LCRs, without implying anything about what particular amino acid composition those LCRs have. This approach emphasizes the patterns of relationships between LCRs, rather than assigned categories of LCRs, which we would argue is fundamentally different from the approach the reviewer suggested.

B) When we refer to LCR type and copy number, we are referring to the copy number of each of the respective types, not the total number of LCRs in the protein. Therefore, determining this requires types to be defined first, which is something that other approaches do not do.

For example, using SEG on G3BP1 and setting an arbitrary threshold here of 0.45 we find 3 SEG-derived LCRs (highlighted in blue here) which correspond to 144-160 ('PQEESEEEVEEPEERQ'), 192-205 ('EPEPDPEPEPEQE'), and 430-448 ('PGGPRGGLGGGMRGPPRG').The authors' approach identifies basically the same three regions:SEG: PQEESEEEVEEPEERQDotMatrix: TEPQEESEEEVEEPEERQQSEG: EPEPDPEPEPEQEDotMatrix: AEPEPDPEPEPEQEPVSSEG: PGGPRGGLGGGMRGPPRGDotMatrix: LRGPGGPRGGLGGGMRGPPRGGMNow I want to be clear – I have no particular love for SEG, but my point is simply that to claim there is no way to characterize LCRs that takes type and copy number is demonstrably false.

We appreciate the reviewer taking the time to perform this analysis. We will address the broader points of the reviewer in our response to the next comment, but we would first like to respond specifically to the reviewer's analysis.

All we intend to claim is that the current approaches for calling LCRs cannot do what dotplot matrices do, which is to (1) call LCRs, *and* (2) Determine LCR relationships and their respective copy numbers. It is unclear how the analysis provided by the reviewer contradicts this claim (i.e. demonstrates that SEG can determine LCR type and copy number). The reviewer has shown that SEG can call LCRs. SEG alone cannot determine the sequence relationships of these LCRs. The LCRs called by SEG can definitely be used for subsequent analysis to determine their sequence relationships. However, this is not part of the SEG method.

Furthermore, above we explain why our approach for determining LCR type and copy number is fundamentally different from the approach of assigning categories and counting.

I would encourage the authors to not oversell the method as some magic bullet, and not pretend one could not do what they have done here with another method. You could.

We do not intend to claim that it is not possible to do what we have done without dotplot matrices. All we wish to claim is that the current approaches for calling LCRs cannot do what dotplot matrices do, which is to (1) call LCRs, *and* (2) Determine LCR relationships and their respective copy numbers. However, we’ve come to realize that there are some regions of the text which inadvertently suggest this, including the sentence:

'As a consequence, we have not been able to systematically understand how LCR sequence and organization influence their function.'

This is an example of where we agree with the reviewer. This sentence suggests that the outputs of the current approaches could not have been used to achieve this even via other downstream analyses, which is incorrect. We have changed the wording of this section so that it only comments on the abilities of the current approaches, and not what their outputs can be used for.

We’d like to emphasize that the overall goal of the manuscript is not to compare our dotplot method to existing methods, but to show that by providing a unified view of the sequences, features, and relationships of LCRs, our approach facilitates new discoveries about how LCRs contribute to higher order assemblies, and provides a framework for understanding the role of LCRs across sequence space.

The big difference is people haven't and the actual analysis of proteomes is interesting and I think well written and well-explored.

This is our primary goal. We wish to make the argument that (1) Type and copy number analysis is something that can potentially be fruitful since it plays important roles in cases where it has been studied, and (2) dotplot matrices provide *one* way to achieve this in a straightforward way.

Codon bias remains undiscussedWhen the authors discuss biases and bridging among distinct classes of LCRs, I would, again, STRONGLY encourage them to consider codon biases – can overlapping/related enrichments for specific residues be explained based on codon distance in repetitive and slippy sequences? I recognize the authors state that 'this is out of scope' but… it's not. If the authors are going to find LCRs with specific compositional biases and comment on bridges between distinct classes of LCRs, ignoring a codon-based explanation for these bridges is most definitely within scope.

We appreciate the reviewer reiterating and trying to clarify the relevance of codon bias here, but respectfully, we are not sure exactly what you mean by codon bias, which we interpret as relative codon frequencies for synonymous codons. Assuming you are not referring to this, and instead you are explicitly referring to the "codon distance in repetitive and slippy sequences", we are still unclear on how codon bias is within the scope of this paper (even though we absolutely agree that it is a related topic and very interesting). We acknowledge that we may be missing something, but we ask the reviewer to consider our reasoning for why we believe it is outside the scope, as follows:

The hypothesis that the reviewer is proposing, as we understand it, is that

1) If the amino acids which make up certain LCR bridges are encoded by codons close in codon distance (i.e. 'AGT' for Serine to 'AGA' for Arginine), then

2) this may be a sufficient explanation for the existence of these bridges,

and so

3) the sequences in the bridge are unlikely to be functional.

While we agree that conceptually this line of reasoning is valid and that it is often used in the absence of functional information, we believe that functional information can be used to test this hypothesis with the following logic: if bridges which correspond to amino acids which are close in codon distance are indeed functional, then these codon similarities do not point to a lack of function. Instead, they may point to interesting hypotheses about how these LCRs arose from ancestral sequences.

As we mention in the section corresponding to Figure 3 and the discussion, there are sequence bridges which are functional (e.g. RS domains for nuclear speckles). This is despite their codons being similar in sequence (Serine – AGT/AGC, Arginine – AGA/AGG). Thus, even if points 1 and 2 are true in the reasoning above (as is the case for RS domains), point 3 does not necessarily follow.

Based on this interpretation of the reviewer's comment, this concern only applies to the question of how LCRs originate, which is not within the scope of the paper. This is not to say that we don't find the question fascinating–these questions were a topic of much discussion when pursuing this project–we just think it would be better suited for further studies which can give this topic a more focused and comprehensive treatment.

LCRs are shortOne thing I had not appreciated in the original manuscript is that the LCRs are short. In the human proteome, 87% of LCRs are 20 residues or shorter. This to me makes me wonder if LCR is really the right term. The LCRs identified previously have generally been large domains, while here LCRs are functionally of the same length as motifs, should these perhaps be defined as LCMs (low complexity motifs) more

It is unclear to us which previously identified LCRs the reviewer is referring to here. In the absence of that, we compared the length distribution of our LCR calls with that of SEG to evaluate the reviewer comment. Based on the results of this comparison, which can be seen in Author response image 3, we believe our approach calls LCRs of reasonable length.

**Author response image 3. sa2fig3:** 

For SEG, 75% of LCRs called are 20 residues or shorter, and the 87th percentile is only at 26 residues in length (see Author response image 3). We do not find this different enough from the LCR lengths called using dotplots to justify a different classification for the sequences identified with our approach, especially given that SEG fuses LCRs in proximity to each other and as such may result in a length distribution shifted toward longer sequences (as we discussed in our response to the previous round of review).